# Spiking Vision Transformer with Saccadic Attention

**Shuai Wang**[1], **Malu Zhang**[1]*, **Dehao Zhang**[1], **Ammar Belatreche**[2], **Yichen Xiao**[1],
**Yu Liang**[1], **Yimeng Shan**[3], **Qian Sun**[1], **Enqi Zhang**[1], **Yang Yang**[1]

[1]University of Electronic Science and Technology of China
[2]Northumbria University, [3]Liaoning Technical University

## Abstract

The combination of Spiking Neural Networks (SNNs) and Vision Transformers (ViTs) holds potential for achieving both energy efficiency and high performance, particularly suitable for edge vision applications. However, a significant performance gap still exists between SNN-based ViTs and their ANN counterparts. Here, we first analyze why SNN-based ViTs suffer from limited performance and identify a mismatch between the vanilla self-attention mechanism and spatio-temporal spike trains. This mismatch results in degraded spatial relevance and limited temporal interactions. To address these issues, we draw inspiration from biological saccadic attention mechanisms and introduce an innovative Saccadic Spike Self-Attention (SSSA) method. Specifically, in the spatial domain, SSSA employs a novel spike distribution-based method to effectively assess the relevance between Query and Key pairs in SNN-based ViTs. Temporally, SSSA employs a saccadic interaction module that dynamically focuses on selected visual areas at each timestep and significantly enhances whole scene understanding through temporal interactions. Building on the SSSA mechanism, we develop a SNN-based Vision Transformer (SNN-ViT). Extensive experiments across various visual tasks demonstrate that SNN-ViT achieves state-of-the-art performance with linear computational complexity. The effectiveness and efficiency of the SNN-ViT highlight its potential for power-critical edge vision applications.

## 1 Introduction

Vision Transformers (ViTs) (Dosovitskiy, 2020) revolutionize the traditional computer vision field, achieving higher performance in many vision tasks such as image classification (Chen et al., 2021; Han et al., 2023) and object detection (Fang et al., 2021c; Touvron et al., 2021). However, ViTs always demand significant computational and memory resources, which greatly restricts their deployment in resource-constrained edge vision environments (Wu et al., 2022; Graham et al., 2021). Consequently, the development of energy-efficient and high-performance solutions remains a significant area of research that necessitates further investigation (Cai et al., 2019; Han et al., 2020b).

Spiking Neural Networks (SNNs), as the third generation of neural networks (Maass, 1997; Gerstner & Kistler, 2002; Izhikevich, 2003; Masquelier et al., 2008), mimics biological information transmission mechanisms using discrete spikes as the medium for information exchange. Spiking neurons fire spikes only upon activation and remain silent at other times. This event-driven mechanism (Caviglia et al., 2014) promotes sparse synapse operations and avoids multiply-accumulate (MAC) operations, which significantly boost the energy efficiency of these models (Zhang et al., 2023). However, the architectures of most SNN-based models still revolve around traditional structures such as CNNs (Fang et al., 2021b; Xing et al., 2019) and ResNets (Fang et al., 2021a; Hu et al., 2024), which exhibit a significant performance gap compared to ViTs.

In recent years, numerous researchers have dedicated efforts to develop SNN-based ViT models. However, most studies (Zhou et al., 2023b; Wang et al., 2023b) retain energy-intensive MAC operations in self-attention computational paradigm and not fully take advantage of SNNs' energy

---

*Corresponding author: maluzhang@uestc.edu.cn

efficiency. Furthermore, these approaches still rely on the Dot-Product operation to measure the spatial relevance between Query ($Q$) and Key ($K$) pairs. However, they fail to account for whether the Dot-Product is well-suited to the binary spike characteristics of SNNs. Subsequently, inspired by Metaformer (Yu et al., 2023), Spike-driven V2 (Yao et al., 2024b) introduces a MAC-free method, and SpikingResformer (Shi et al., 2024) combines ResNet-based architecture and self-attention computation paradigm to further reduce parameters. These methods ensure the high performance of SNN-based ViTs while achieving a full spike-driven manner, offering significant energy savings. Nevertheless, these studies treat self-attention computational paradigm merely as an efficient token mixer (Yu et al., 2022), without exploring an effective paradigm suited to spike trains. Furthermore, these methods primarily focus on spatial feature extraction, overlooking the temporal dynamics of SNNs. Consequently, exploring spiking self-attention paradigms tailored to the spatio-temporal characteristic of SNNs represents a potential area for improvement.

Biological vision dynamically captures and understands visual scenes through saccadic mechanisms (Melcher & Morrone, 2003; Binda & Morrone, 2018; Guadron et al., 2022). It focuses on specific visual areas at each moment and utilizes dynamic saccadic movements across the temporal domain to achieve a contextual understanding of the entire visual scene (Hanning et al., 2023). Compared to vanilla self-attention mechanisms (Liu et al., 2021b), it offers higher energy and computational efficiency. Additionally, the saccadic process involves intense temporal interactions (Idrees et al., 2020), which closely align with the unique temporal characteristics of SNNs. Therefore, we draw inspiration from the saccadic mechanisms to design a Saccadic Spike Self-Attention (SSSA) method. The SSSA method adapts to the spatio-temporal characteristics of SNNs, enabling an efficient and effective comprehensive understanding of visual scenes. Based on this, we further develop a SNN-based Saccadic Vision Transformer. The summary contributions are as follows:

- We thoroughly analyze the reasons for the mismatch between the vanilla self-attention mechanism and SNNs. In the spatial domain, the binary and sparse nature of spikes creates significant magnitude differences between $Q$ and $K$ in SNN-based ViTs, making it difficult for vanilla self-attention to assess spatial relevance. Additionally, vanilla self-attention is designed for ANNs and neglects the temporal interactions among timesteps in SNNs, limiting its ability to explore information in the temporal domain.

- We propose a Saccadic Spike Self-Attention (SSSA) mechanism specifically designed for SNNs' spatio-temporal characteristics. In the spatial domain, SSSA introduces a novel spike distribution-based method to measure relevance between $Q$ and $K$ pairs effectively. Temporally, SSSA introduces a saccadic interaction module that dynamically focuses on selected visual areas and achieves a comprehensive understanding of the whole scene.

- To further enhance the computational efficiency of SSSA, we introduce a linear complexity version called SSSA-V2. It is mathematically linear scaling mapping to SSSA, preserving all performance benefits. Additionally, SSSA-V2 successfully reduces computational complexity to a linear level and works in a fully event-driven manner.

- Building on the proposed SSSA mechanisms, we develop a SNN-based Vision Transformer (SNN-ViT) architecture. Extensive experiments are conducted on various visual tasks demonstrating that SNN-ViT achieves SOTA performance with linear computational complexity. It presents a promising approach for achieving both high-performance and energy-efficient visual solutions.

## 2 RELATED WORK

**Vision Transformers:** ViTs segment images into patches and apply self-attention (Vaswani, 2017; Kenton & Toutanova, 2019) to learn inter-patch relationships, outperforming CNNs across multiple vision tasks (Mei et al., 2021; Bertasius et al., 2021; Guo et al., 2021). Nevertheless, ViTs face challenges like high parameter counts (Liu et al., 2021b), and increased computational complexity proportional to token length (Pan et al., 2020; Liu et al., 2022). To enhance the computational efficiency of ViTs, many researchers (Jie & Deng, 2023; Li et al., 2023) are focused on exploring lightweight improvement methods. For example, LeViT (Graham et al., 2021) incorporates convolutional elements to expedite processing, and MobileViT (Mehta & Rastegari, 2021) combines lightweight MobileNet blocks with MHSA, achieving lightweight ViTs successfully. However, these enhance-

ments still rely on expensive MAC computations which are not suitable for resource-limited devices. This highlights the need for investigating more energy-efficient ViT solutions.

**Spiking Neural Networks:** The event-driven mechanism enhances the energy efficiency of SNNs, offering a significant advantage for compute-constrained edge devices. With the introduction of ANN-SNN (Cao et al., 2015; Han et al., 2020a; Wu et al., 2021) and direct training (Wu et al., 2018; Fang et al., 2021b; Zhang et al., 2021; Wei et al., 2023) algorithm, the difficult associated with training high-performance SNNs is significantly reduced. Based on these advanced learning algorithms, some research (Hu et al., 2021; Zheng et al., 2021; Hu et al., 2024) propose deep residual SNNs (Wang et al., 2024; Shan et al., 2024) and others (Yao et al., 2023; Zhu et al., 2024; Shan et al., 2024) contribute multi-dimensional spike attention mechanisms, achieving competitive performance on many tasks (Zhang et al., 2024). These improvements further enhance the application of SNNs in various visual tasks. However, despite rapid advancements, a significant performance gap remains between these traditional deep SNN architectures and the latest ViTs.

**Vision Transformers Meet Spiking Neural Networks:** To explore high-performance and energy-efficient visual solutions, SNN-based ViTs (Zhou et al., 2023b; Wang et al., 2023a) have emerged. Spikformer (Zhou et al., 2023b;a) pioneers a spike-based self-attention computation, establishing the first spiking ViT. However, they still utilize expensive MAC operations and matrix multiplication in self-attention computation, which are inefficient for binary spikes. Recently, Spike-driven Transformer (Yao et al., 2024b) implements Hadamard product in the self-attention module for a fully spike-driven ViT. Additionally, SpikingResformer (Shi et al., 2024) integrates a Dual Spike self-attention module for improved performance and energy efficiency. However, these models primarily treat self-attention as a token mixer (Yu et al., 2022), without exploring an effective relevance computation suited to spike trains. Moreover, they also overlook the temporal dynamics of SNNs. (Zhang et al., 2021; Bohte et al., 2000). Therefore, developing spike self-attention mechanisms tailored to the spatio-temporal characteristics of SNNs is essential for further advancements.

## 3 PROBLEM ANALYSIS IN SPIKING SELF-ATTENTION

In this section, we analyze the mismatches between vanilla self-attention mechanisms and SNNs in both the spatial and temporal domains. The detailed discussion is provided in the following sections.

### 3.1 DEGRADED SPATIAL RELEVANCE

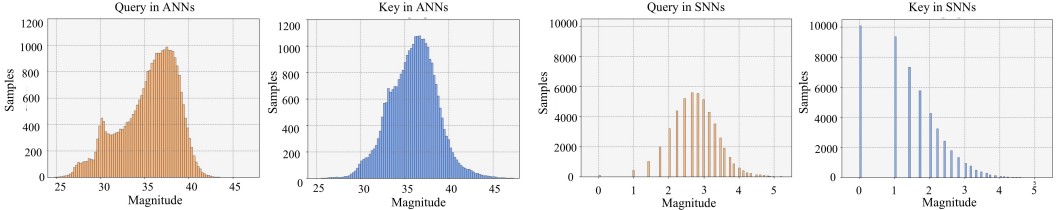

Figure 1: Distribution of magnitudes for Q and K in ViTs within ANNs and SNNs on CIRAF100. In ANNs, Q and K exhibit similar magnitude distributions, whereas in SNNs, the magnitude differences between Q and K are pronounced.

The vanilla self-attention measures the spatial relevance between $Q$ and $K$ through Dot-Product operation. For a given query $Q_i$ and key $K_i$ vector, the relevance between them are as follows:

$$\text{Dot-Product}\left(\mathcal{Q}_i, \mathcal{K}_i\right) = \sum_{j=1}^{D} \mathcal{Q}_{ij}\mathcal{K}_{ij}, \tag{1}$$

$D$ is the dimension of both vectors, $Q_{ij}$ and $K_{ij}$ refer to the $j$-th elements of these vectors, respectively. Notably, the relevance based on Dot-Product takes into account both the angle and magnitude of the vectors (Kim et al., 2021). When there is a significant difference in magnitude between vectors, the Dot-Product may not accurately measure their spatial relevance.

In ANNs, continuous input $X$ is first normalized using layer normalization (Dosovitskiy, 2020) and then be processed through linear transformations $W_Q$ and $W_K$ to derive the matrices $Q$ and $K$. This ensures that the magnitudes of $Q$ and $K$ are closely matched (Xu et al., 2019), preventing large variations between vectors. As shown in the left part of Fig.1, the distribution between $Q$ and $K$ across various datasets remains nearly identical, allowing effective measuring of the spatial relevance for attention score in ANNs.

Due to the discrete activation characteristics of spiking neurons, the continuous distribution of the normalized membrane potentials in $Q$ and $K$ is disrupted. As shown in the right part of Fig. 1, the magnitude of $Q$ and $K$ in SNNs shows significant variability, which leads to the failure of the Dot-Product in measuring spatial relevance. Moreover, despite $Q$ and $K$ following identical distributions, the sparsity of binary spikes significantly reduces their stability compared to ANNs. We provide a detailed analysis of this assertion in Appendix. A. Therefore, developing more effective methods to measure the spatial relevance between spike trains could be a viable approach to enhancing the performance of SNN-based ViTs.

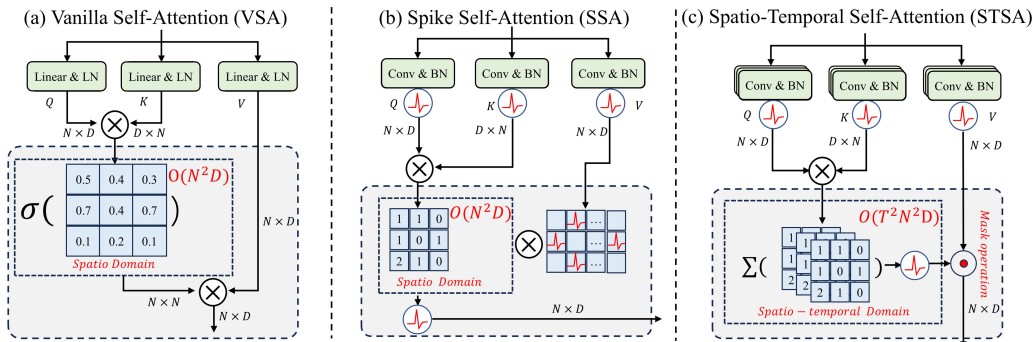

Figure 2: Comparison of three self-attention computation paradigms. (a) VSA employs floating-point matrix multiplication to assess the spatial correlation between Q and K, resulting in a computational complexity of $\mathcal{O}(N^2D)$. (b) SSA lacks a dedicated temporal interaction module, maintaining the same complexity as VSA. (c) In contrast, STSA introduces global spatial-temporal interactions, increasing the complexity to $\mathcal{O}(T^2N^2D)$.

## 3.2 LIMITED TEMPORAL INTERACTION

As shown in Fig. 2(a), vanilla self-attention in ViTs operates independently of timesteps, thereby preventing the need for temporal interaction in self-attention designs. Conversely, SNNs rely on multiple timesteps to enrich their information representation capabilities (Fang et al., 2021b). However, as shown in Fig. 2(b), most spike self-attention mechanisms (Yao et al., 2024b; Zhou et al., 2023b; Shi et al., 2024) lack dedicated modules for the temporal domain. The only temporal interaction in those methods is the accumulation of historical information by spiking neurons (LIF neurons), whose dynamics can be described as:

$$U[t + 1] = H[t] + X[t + 1], \tag{2}$$
$$S[t + 1] = \Theta(U[t + 1] - V_{th}), \tag{3}$$
$$H[t + 1] = V_{reset}S[t + 1] + \tau U[t + 1](1 - S[t + 1]). \tag{4}$$

$X[t + 1]$ denotes the spatial input current, while $H[t]$ and $U[t]$ represent the pre-synaptic and post-synaptic membrane potentials, respectively. The Heaviside function $\Theta(\cdot)$ is employed for spike generation. If a spike occurs ($S[t + 1] = 1$), $H[t]$ resets to $V_{reset}$; otherwise, $U[t + 1]$ decays with a time constant $\tau$ and feeds into $H[t + 1]$. However, due to the reset and decay mechanism, the residual membrane potential cannot sustain long-range dependencies, resulting in a significant loss of historical information. To solve this problem, (Wang et al., 2023b) proposes a spatio-temporal spike self-attention method as shown in Fig. 2(c). But this method has $\mathcal{O}(T^2N^2D)$ computational complexity, significantly restricting the training efficiency of SNNs and increasing the complexity of deployment. Therefore, achieving more effective spatio-temporal interactions without increasing computational overhead remains a pressing challenge.

## 4 SACCADIC SPIKING SELF-ATTENTION MECHANISM

We introduce a Saccadic Spiking Self-Attention (SSSA) method tailored for the spatio-temporal characteristic of SNNs. Spatially, SSSA enhances relevance measurement between spike vectors $Q$ and $K$ based on their distribution forms. Temporally, it incorporates a dedicated saccadic interaction module for dynamic contextual comprehension of the visual scene. Additionally, we advance SSSA to version V2, which retains the high performance of SSSA and reduces computational complexity from $\mathcal{O}(N^2)$ to $\mathcal{O}(D)$.

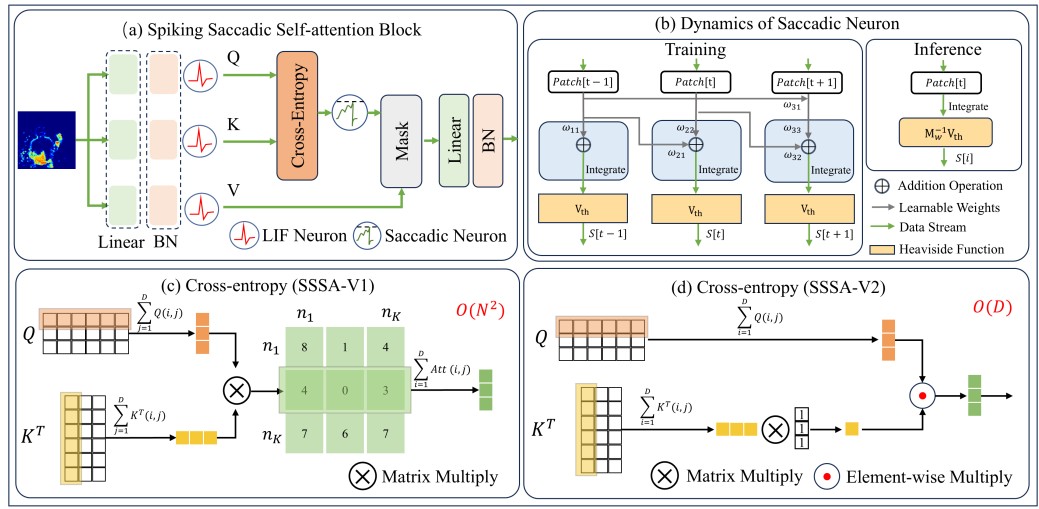

Figure 3: Overview of SSSA method. (a) SSSA consisting of two key components: cross-entropy relevance computation and saccadic spiking neurons. The latter outputs spike-driven decisions that mask $V$ in N-dimensional space. (b) training and inference process for saccadic spiking neurons. (c) the structure of the spatial relevance computation based on spike distribution. (d) the structure of SSSA-V2 on spatial relevance computation, significantly reducing computational complexity.

### 4.1 SPATIAL RELEVANCE COMPUTATION FROM SPIKE DISTRIBUTION

To mitigate the issue of degraded spatial relevance caused by Dot-Product operations, we introduce a novel distribution-based approach. It directly measures the relevance between two vectors using cross-entropy, unaffected by their magnitudes. Further details can be found in Appendix B.

For a patch $\mathbf{x} \in \mathbb{R}^D$ in either $Q$ or $K$, it can be treated as a $D$-dimensional $\{0, 1\}$ random spike train, where $p$ represents the spike firing rate. The cross-entropy between patches $\mathbf{q} \in Q$ and $\mathbf{k} \in K$ is given by:

$$\mathcal{H}(\mathbf{q}, \mathbf{k}) = -\left[p_{\mathbf{q}} \log p_{\mathbf{k}} + (1 - p_{\mathbf{q}}) \log (1 - p_{\mathbf{k}})\right], \tag{5}$$

where $p_{\mathbf{q}}$ and $p_{\mathbf{k}}$ denote the firing rates for vectors $\mathbf{q}$ and $\mathbf{k}$, respectively. Since both $\mathbf{q}$ and $\mathbf{k}$ are spike trains, our focus shifts to the distribution of spikes rather than silent states. Consequently, we primarily consider the first term of Eq. 5, allowing us to simplify $\mathcal{H}(q, k)$ to $-p_{\mathbf{q}} \log p_{\mathbf{k}}$. Given that both $\log(x)$ and $x$ maintain the same monotonicity, substituting $\log(x)$ with $x$ is a feasible simplification that preserves the effectiveness of $\mathcal{H}(\mathbf{q}, \mathbf{k})$, while avoiding nonlinear computations. Detailed analysis is provided in Appendix B.

Since cross-entropy $\mathcal{H}(\mathbf{q}, \mathbf{k})$ measures negative relevance, we take its negative as our attention result. As a result, the cross-attention between $Q$ and $K$, denoted as $\mathrm{CroAtt}(Q, K) = -\mathcal{H}(Q, K)$, can be further expressed as:

$$\mathrm{CroAtt}(Q, K) = \mathcal{Q}'\mathcal{K}'^T, \ \mathcal{Q}' = \sum^D Q, \ \mathcal{K}' = \sum^D K, \ Q, K \in \mathbb{R}^{T \times N \times D}. \tag{6}$$

As illustrated in Fig. 2(c), $\mathcal{Q}'$ and $\mathcal{K}'$ represent the sum of spikes across the dimension $D$. This approximation allows for more efficient parallel computation of spatial relevance between $Q$ and $K$. By employing this distribution-based method, we more accurately assess the relevance between vectors with non-standard distributions, thereby addressing the issue of degraded spatial relevance.

## 4.2 SACCADIC TEMPORAL INTERACTION FOR ATTENTION

Biological saccadic mechanisms do not process all visual information at once. Instead, they progressively focus on key visual areas within a scene Guadron et al. (2022). This ensures that biological systems can efficiently achieve contextual understanding of the entire visual scene. Inspired by this mechanism, we have designed an effective temporal interaction module that incorporates two critical processes: salient patch selection and saccadic context comprehension. The first process selectively computes only a subset of patches at each timestep, while ignoring the others. It can significantly reduce the computational complexity of the SSSA method. This process can be described as:

$$\mathcal{P}atch = \sum_{j=1}^{n} \text{CroAtt}\left(Q, K\right), \ \text{CroAtt}\left(Q, K\right) \in \mathbb{R}^{T \times N \times N}, \tag{7}$$

CroAtt$(Q, K)$ represents the spatial relevance between patches in $Q$ and $K$. By summing the rows of the CroAtt$(Q, K)$ matrix, the $\mathcal{P}atch$ represents the spatial salience of patches. Subsequently, the saccadic interaction module makes contextual understanding based on $\mathcal{P}atch$. To ensure the asynchronous characteristics of SNNs, we aim to integrate the interaction process into spiking neurons. However, the significant historical forgetting caused by the resetting and decay mechanism of LIF neurons prevents efficient interaction. Therefore, we introduce a plug-and-play saccadic spiking neuron, whose dynamic during training and inference phases can be described as follows:

$$\text{Training} \begin{cases} \mathbf{H} = \mathbf{M}_w \mathcal{P}\mathbf{atch} \\ \mathbf{S} = \Theta\left(\mathbf{H} - \mathbf{V}_{th}\right) \end{cases} \quad \text{Inference} \begin{cases} H[t] = \mathcal{P}atch[t] \\ S[t] = \Theta\left(H[t] - \mathbf{M}_w^{-1} V_{th}[t]\right) \end{cases} \tag{8}$$

Here, $\mathbf{H}, \mathbf{S}, \mathcal{P}\mathbf{atch} \in \mathbb{R}^{T \times N}$ represents the data format for parallel training, encompassing the entire temporal dimension. $\mathbf{M}_w$ is a learnable lower triangular matrix that precisely regulates contributions from each timestep, facilitating efficient temporal interactions. Utilizing $\mathbf{M}_w$ to compute membrane potentials, saccadic spiking neurons avoid decay or resetting disruptions. As shown in Fig.3, we depict the dynamic process of saccadic spiking neurons. During training, the membrane potential of saccadic spiking neurons is represented as $\sum_0^t w_{it} \mathcal{P}atch[t], w_{it} \in \mathbf{M}_w$. However, all timesteps are processed simultaneously via matrix multiplication, which requires substantial computational resources. To maintain SNNs' energy efficiency, we propose an asynchronous inference decoupling method. By incorporating the inverse of $\mathbf{M}_w$ into the threshold levels of the saccadic spiking neurons, we ensure temporal decoupling between $\mathbf{H}$ and $\mathbf{S}$. The spike firing process depends solely on the current values of $H[t]$ and $V_{th} M_w^{-1}[t]$, eliminating the need for historical information. Thus, saccadic spiking neurons ensure the capability for asynchronous inference. Notably, the temporal complexity of saccadic spiking neurons is only $\mathcal{O}\left(T\right)$, significantly superior to the $\mathcal{O}\left(T^2\right)$. The dynamics of saccadic spiking neurons are detailed in Appendix.C.

## 4.3 LINEAR COMPLEXITY AND SPIKE-DRIVEN COMPUTATION

Building on the aforementioned components, SSSA is specifically designed for the spatio-temporal characteristics of SNNs. It enables a more effective comprehensive understanding of the entire visual scene with lower time complexity. Its formulation is described as follows:

$$\text{SSSA}\left(\mathcal{Q}, \mathcal{K}, \mathcal{V}\right) = \Theta\left(\mathbf{M}_w \mathcal{P}atch[0, \cdots, t] - \mathcal{V}_{th}\right) \cdot \mathcal{V} = \Theta\left(\mathbf{M}_w \left(\mathcal{Q}' \times \mathcal{K}'^T\right) L - \mathbf{V}_{th}\right) \cdot \mathcal{V}, \tag{9}$$

where $L$ represents a column vector $[1, 1, \ldots, 1]$ with dimension $N$, facilitating the summation of row elements. However, as depicted in Fig. 2(c), SSSA includes integer multiplication operations within $\mathcal{Q}' \times \mathcal{K}'$, compromising the energy efficiency of the SNNs. Moreover, the quadratic complexity of $\mathcal{Q}' \times \mathcal{K}'$ still indicates potential for optimization. Given that the matrix multiplications in Eq.9 do not involve nonlinear operations, they allow for free association of matrices without altering the computational sequence. Consequently, to avoid the need for integer multiplication and further reduce computational complexity, we conduct an linear scaling mapping of Eq.9, which can be described as follows:

$$\text{SSSA}\left(\mathcal{Q}, \mathcal{K}, \mathcal{V}\right) = \Theta\left(\left(\mathbf{M}_w \times \mathcal{Q}'\right)\left(\mathcal{K}'^T \times L\right) - V_{th}\right) \cdot \mathcal{V} \tag{10}$$

In SSSA-V2, computations begin with the calculation of $Q'$ and $K'$ based on $Q$ and $K$, each with a complexity of $\mathcal{O}(D)$. Then, instead of calculating $Q' \times K'$, SSSA-V2 treats $(\mathcal{K}'^T \times L)$ as a learnable scaling factor $\alpha$, applied to the threshold $\mathbf{V}_{th}$ of the saccadic neuron. Subsequently, $\mathbf{M}_w \times Q'$ as $\mathcal{P}atch[i]$ input into the saccadic neurons. During the inference process, $\mathbf{M}_w$ can be integrated into the thresholds of saccadic neurons to maintain a fully spike-driven system.

$$\text{Inference} \begin{cases} H[t] = \mathcal{Q}'[t], \\ S[t] = \Theta \left( \mathbf{H}[t] - \frac{1}{\alpha} \left( \mathbf{M}_w^{-1} \mathbf{V}_{th} \right) [t] \right). \end{cases} \tag{11}$$

Mathematically, SSSA-V2 is linear scaling mapping to SSSA, preserving all the advantages of SSSA while significantly reducing the need for integer multiplication operations. Additionally, SSSA-V2 achieves a linear spike self-attention mechanism with total computational complexity of $\mathcal{O}(2D+N)$, offering significant benefits in resource-constrained environments.

## 5 SNN-BASED SACCADIC VISION TRANSFORMER

As illustrated in Fig.4, we introduce a novel SNN-ViT based on the proposed SSSA method. It primarily consists of GL-SPS blocks and SSSA-based transformer blocks. The following section will provide detailed descriptions of these components.

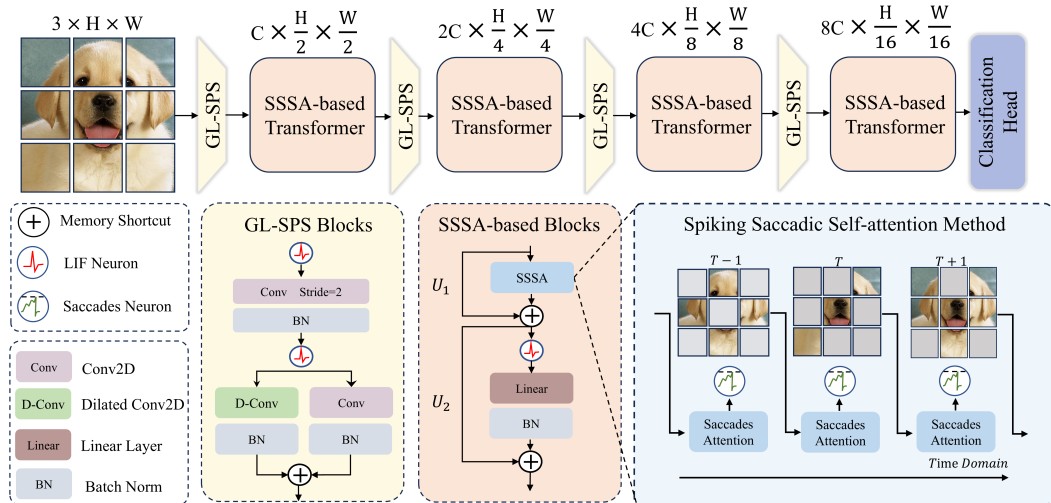

Figure 4: The overall structure of SNN-ViT, mainly consisting of GL-SPS blocks and SSSA-based transformer blocks. GL-SPS block combines dilated convolution and standard convolution at different scales to facilitate multi-scale feature extraction from images. The SSSA-based block, composed of SSSA methods and Linear layers, achieves lower computational complexity.

### 5.1 GL-SPS: GLOBAL-LOCAL SPIKING PATCH SPLITTING MODULE

Currently, existing SPS methods primarily rely on shallow spiking convolution modules to capture local information, which prevents the effective extraction of multi-scale features. This limitation leads to degraded performance in processing wide-field image features. To address this issue, we design the Global-Local convolutional SPS block, described as follows:

$$\text{GL-SPS}\left( X[t] \right) = \text{BN}\left( \text{Conv}(X[t]) + \text{BN}\left( \text{DConv}\left( X[t] \right) \right) \right), \tag{12}$$

where $X[t]$ is the result of a convolution operation with a stride of 2. Conv2d and D-Conv2d represents standard and dilated convolution operations. $\text{BN}(\cdot)$ is Batch Normalization. The GL-SPS utilizes both Conv2d and D-Conv2d to extract features. Combining layers with different dilation rates effectively gathers context from various visual scales. Consequently, SNN-ViT employs the GL-SPS method as its embedding module, enhancing efficiency and scalability in feature extraction.

## 5.2 OVERALL ARCHITECTURE

Building upon the pyramid structure (Liu et al., 2021b; Yu et al., 2023), we propose a novel SNN-ViT that incorporates the GL-SPS block and the SSSA method. GL-SPS part encodes the input image through downsampling operation and various convolutions operation. Specifically, the downsampling operation is defined as a convolution operation with a kernel size of 7 and a stride of 2. The GL-SPS method follows the previous section. The whole block is defined as follows:

$$U_0 = \text{GL-SPS}(I) \qquad\qquad I \in \mathbb{R}^{T \times C \times H \times W} \qquad (13)$$

Subsequently, $U_0$ is inputted into the SSSA-based block, which consists of SSSA method and MLP Layer. To further reduce the computational complexity of the model, we adopt the SSSA-V2 version as the paradigm for self-attention computation in the architecture. Subsequently, the output from the SSSA-based Transformer blocks is fed into the Global Average Pooling (GAP) module, followed by a Classification Head (FCH) that generates the prediction Y. These parts can be defined as:

$$U_1 = U_0 + \text{BN}(\text{Conv}([\text{SSSA}(\mathcal{SN}(U_0))])), \qquad U_0 \in \mathbb{R}^{T \times N \times D} \qquad (14)$$
$$U_2 = U_1 + \text{BN}(\text{Linear}[\mathcal{SN}(U_1)]), \qquad U_1 \in \mathbb{R}^{T \times N \times D} \qquad (15)$$
$$Y = \text{FCH}(\text{GAP}(\mathcal{SN}(U_2))), \qquad\qquad (16)$$

where $Y$ denotes the predicted outcome. For different types of datasets, we can integrate the GL-SPS component with varying numbers of SSSA decoding blocks. The details of the network architecture and the parameter count are presented in Appendix.F.

## 6 EXPERIMENTS

### 6.1 IMAGE CLASSIFICATION

SNN-ViT is evaluated on both static and neuromorphic datasets, including CIFAR10, CIFAR100 (Krizhevsky et al., 2009), ImageNet (Deng et al., 2009) and CIFAR10-DVS (Li et al., 2017). Specifically, for ImageNet, the input image size is $3 \times 224 \times 224$, with batch sizes of 128, and training epoch is conducted over 310. Our experimental results are summarized in Table.1 and 2.

Table 1: Summary of Network Architectures and Parameters across Vary Datasets

| Method | CIFAR10 | | CIFAR100 | | CIFAR10-DVS | |
|---|---|---|---|---|---|---|
| | Param. | Acc. | Param. | Acc. | Param. | Acc. |
| PLIF (Fang et al., 2021b) | - | 93.50 | - | - | - | 74.8 |
| tdBN (Zheng et al., 2021) | - | 93.2 | - | - | - | 67.8 |
| DSpike (Li et al., 2021) | - | 94.25 | - | 74.24 | - | 75.4 |
| TET (Deng et al., 2022) | 12.63 | 94.44 | 12.63 | 74.47 | - | 77.33 |
| Spikformer (Zhou et al., 2023b) | 9.32 | 95.51 | 9.32 | 78.21 | 2.57 | 80.9 |
| Spikingformer (Zhou et al., 2023a) | 9.32 | 95.81 | 9.32 | 78.21 | 2.57 | 81.3 |
| Spike-driven (Yao et al., 2024b) | 10.28 | 95.60 | 10.28 | 78.40 | 2.57 | 80.0 |
| STSA (Wang et al., 2023b) | - | - | - | - | 1.99 | 79.9 |
| **SNN-ViT (Ours)** | **5.57** | **96.1** | **5.57** | **80.1** | **1.52** | **82.3** |

To facilitate a comprehensive comparison with similar works, we meticulously documented the performance of our SNN-ViT across varying model sizes. In the CIFAR100 dataset, the direct training performance of SNN-ViT significantly surpasses the Spike-driven Transformer (Yao et al., 2024b) with fewer parameters. This underscores the high robustness and sensitivity of the SSSA strategy in smaller-scale tasks, closely mirroring biological cognitive processes. Notably, for neuromorphic datasets, the computational complexity of SNN-ViT is $\mathcal{O}(TD)$, while STSA (Wang et al., 2023b) has a complexity of $\mathcal{O}(T^2 N^2 D)$. Despite reducing computational complexity by three orders, SNN-ViT maintains SOTA performance. Furthermore, on the ImageNet-1K dataset, we conduct a detailed comparison of SNN-ViT with other similar works. The comparison focuses on computational complexity, parameter count, energy consumption, and accuracy. Experimental results demonstrate that SNN-ViT achieves SOTA performance under linear computational complexity.

Table 2: Detailed comparison with other similar methods on ImageNet-1K

| Method | Architecture | Complexity | Time Step | Param. (M) | Energy (mJ) | Acc. (%) |
|---|---|---|---|---|---|---|
| ViT | ViT-12-768 | $\mathcal{O}(N^2D)$ | - | 86M | 80.86 | 77.90 |
| (Dosovitskiy, 2020) | ViT-24-1024 | $\mathcal{O}(N^2D)$ | - | 307M | 283.36 | 76.51 |
| Swin Transformer | Swin-T | $\mathcal{O}(ND^2)$ | - | 29M | 20.72 | 81.35 |
| (Liu et al., 2021b) | Swin-S | $\mathcal{O}(ND^2)$ | - | 51M | 40.24 | 83.03 |
| Flatten Transformer | FLatten-Swin-T | $\mathcal{O}(N^2D)$ | - | 29M | 20.72 | 82.14 |
| (Han et al., 2023) | FLatten-Swin-S | $\mathcal{O}(N^2D)$ | - | 51M | 40.24 | 83.52 |
| Spikformer | Spikformer-8-384 | $\mathcal{O}(N^2D)$ | 4 | 16.8M | 7.73 | 70.24 |
| (Zhou et al., 2023b) | Spikformer-8-512 | $\mathcal{O}(N^2D)$ | 4 | 29.7M | 11.6 | 73.38 |
| | Spikformer-8-768 | $\mathcal{O}(N^2D)$ | 4 | 66.3M | 21.5 | 74.81 |
| SpikingResformer | SpikingResformer-S | $\mathcal{O}(N^2D)$ | 4 | 17.8M | 3.37 | 75.95 |
| (Shi et al., 2024) | SpikingResformer-M | $\mathcal{O}(N^2D)$ | 4 | 35.5M | 5.46 | 77.24 |
| | SpikingResformer-L | $\mathcal{O}(N^2D)$ | 4 | 60.4M | 8.76 | 78.77 |
| Spike-driven | Spike-driven-8-384 | $\mathcal{O}(ND)$ | 4 | 16.8M | 3.90 | 72.28 |
| (Yao et al., 2024b) | Spike-driven-8-512 | $\mathcal{O}(ND)$ | 4 | 29.7M | 4.50 | 74.57 |
| Meta-SpikeFormer | Meta-SpikeFormer-384 | $\mathcal{O}(ND^2)$ | 4 | 33.1M | 32.8 | 74.10 |
| (Yao et al., 2024a) | Meta-SpikeFormer-512 | $\mathcal{O}(ND^2)$ | 4 | 55.4M | 52.4 | 79.70 |
| **SNN-ViT(Ours)** | SNN-ViT-8-256 | $\mathcal{O}(D)$ | 4 | 13.7M | 14.28 | 74.66 |
| | SNN-ViT-8-384 | $\mathcal{O}(D)$ | 4 | 30.4M | 20.83 | 76.87 |
| | SNN-ViT-8-512 | $\mathcal{O}(D)$ | 4 | 53.7M | 35.75 | **80.23** |

## 6.2 REMOTE OBJECT DETECTION

Given the high adaptability of biological saccadic mechanisms to dynamic visual tasks, we aim to apply SNN-ViT to object detection tasks to demonstrate its advantages. As SNNs are often employed in resource-constrained edge computing scenarios, we select two remote sensing datasets: NWPU VHR-10 Cheng et al. (2017) and SSDD (Wang et al., 2019). The NWPU VHR-10 dataset comprises very high-resolution (VHR) images across ten categories, including airplanes, ships, storage tanks, baseball diamonds, tennis courts, basketball courts, ground track fields, harbors, bridges, and vehicles. The SSDD dataset focuses on ship detection using Synthetic Aperture Radar (SAR)

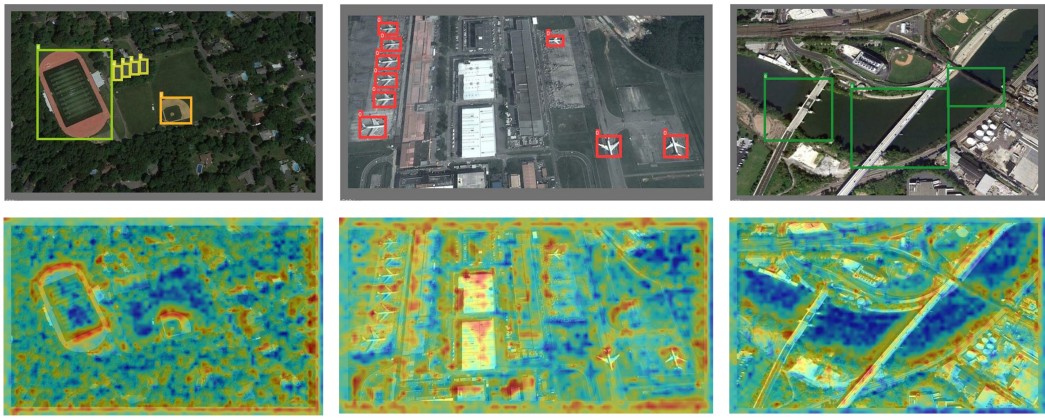

Figure 5: The detection results of SNN-ViT-YOLO on the NWPU-10 dataset are displayed in the first rows. SSSA attention heatmaps are showcased in the second rows.

images. We build a SNN-ViT-YOLO framework by incorporating the SNN-ViT as the backbone of the YOLO-v3 architecture. The structure details are shown in Appendix.G. As shown in Table.3,

to validate the superior performance of SSSA in dynamic visual tasks, we present a comparison with various deep ANN-based object detection methods, including YOLO-v3 (Liu et al., 2021a), RICNN (Cheng et al., 2016), and Faster RCNN (Fu et al., 2020). Additionally, we also compare our approach with EMS-YOLO (Su et al., 2023), the current SOTA results in the SNN fields. The results indicate that SNN-ViT-YOLO outperforms other methods on the two datasets. This demonstrates that SNN-ViT offers a viable approach for aviation and satellite image analysis in extreme environments.

Table 3: Performance comparison with ANNs and SNNs on NWPU VHR-10 and SSDD.

| Dataset | Method | Spike-driven | Timestep | mAP@0.5(%) |
|---------|--------|:---:|:---:|:---:|
| NWPU VHR-10 | YOLO-V3 (Liu et al., 2021a) | ✗ | - | 87.3% |
| | $CS^n$Net (Chen et al., 2023) | ✗ | - | 90.4% |
| | EMS-YOLO (Su et al., 2023) | ✓ | 4 | 86.5% |
| | | ✓ | 8 | 87.9% |
| | **SNN-ViT-YOLO (Ours)** | ✓ | 4 | 88.2% |
| | | ✓ | 8 | 89.4% |
| SSDD | Faster R-CNN Fu et al. (2020) | ✗ | - | 85.3% |
| | EMS-YOLO Su et al. (2023) | ✓ | 4 | 94.8% |
| | | ✓ | 8 | 95.1% |
| | **SNN-ViT-YOLO (Ours)** | ✓ | 4 | 96.7% |
| | | ✓ | 8 | 97.0% |

## 6.3 ABLATION STUDY

To verify the effectiveness of each component in the SNN-ViT, we perform a comprehensive ablation study in the CIFAR100 dataset. The Spikformer (Zhou et al., 2023b) is selected as the baseline for comparison. Subsequently, we replace the corresponding modules in the baseline with SSSA blocks and GL-SPS blocks to assess their impact on performance. As shown in Table 4, replacing our SSSA method im-

Table 4: Ablation Study

| Model | Param. (M) | Complexity | Acc. (%) |
|-------|:---:|:---:|:---:|
| Baseline | 5.76 | $\mathcal{O}(N^2D)$ | 76.95 |
| +SSSA | 5.52 | $\mathcal{O}(D)$ | 79.60 + (2.65) |
| +GL-SPS | 5.81 | $\mathcal{O}(N^2D)$ | 77.88 + (0.93) |
| +both | 5.57 | $\mathcal{O}(D)$ | 80.1 + (3.15) |

proves performance by approximately 2.65%, while reducing computational complexity to $\mathcal{O}(N^2)$. Then we also verify the effectiveness of the GL-SPS blocks. As shown in Table.4, GL-SPS blocks achieve a performance improvement of about 0.93% compared to baseline. This further demonstrates the enhanced compatibility of multi-scale feature maps with the saccadic process. Finally, we replace both SSSA and GL-SPS, achieving an approximately 3.15% performance improvement. Ablation studies validate that the SSSA indeed can significantly enhance performance, confirming its compatibility with spatio-temporal spike trains.

## 7 CONCLUSION

This work provides a detailed analysis of the mismatch between the vanilla ViT and spatio-temporal spike trains. This mismatch results in degraded spatial relevance and limited temporal interactions. Inspired by the biological saccadic attention mechanism, this work proposes a SSSA method tailored to the SNNs. In the spatial dimension, SSSA employs a more efficient distribution-based approach to compute the spatial relevance between Query and Key in SNNs. In the temporal domain, SSSA utilizes a dedicated saccadic interaction module, calculating only a subset of patches at each timestep to dynamically understand the context of the entire visual scene. Building on SSSA method, we develop a SNN-ViT structure, which achieves state-of-the-art performance across various visual tasks with linear computational complexity. SNN-ViT effectively integrates advanced biological cognitive mechanisms with artificial intelligence techniques, providing a promising avenue for exploring high-performance, energy-efficient edge visual tasks.

# 8 ACKNOWLEDGMENT

This work was supported in part by the National Natural Science Foundation of China under grants U20B2063, 62220106008, and 62106038, the Sichuan Science and Technology Program under Grant 2024NSFTD0034 and 2023YFG0259, the Open Research Fund of the State Key Laboratory of Brain-Machine Intelligence, Zhejiang University (Grant No.BMI2400020).

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

# A  LIMITATIONS OF DOT-PRODUCT FOR SPIKE TRAINS

The Dot-Product is the operation to measure relevance between two vectors $\mathbf{u}$ and $\mathbf{v}$ in an $n$-dimensional space, which is defined as:

$$\mathbf{u} \cdot \mathbf{v} = \sum_{i=1}^{n} u_i v_i = \|\mathbf{u}\|\|\mathbf{v}\| \cos \theta \tag{17}$$

where $u_i$ and $v_i$ are the components of vectors $\mathbf{u}$ and $\mathbf{v}$ respectively, $\|\mathbf{u}\|$ and $\|\mathbf{v}\|$ denote the magnitudes (norms) of the vectors, and $\theta$ is the angle between them. This expression clearly illustrates that the Dot-Product is influenced by both the magnitudes of the vectors and the cosine of the angle between them. Variations in either magnitude or angle will affect the result of the Dot-Product, thus affecting the measure of relevance between the vectors.

**Problem: If the $Q$ and $K$ are controlled to be similar distributions in SNNs, would the effectiveness of the Dot-Product still be influenced by magnitude differences?**

**Analysis:** To deepen our investigation, we present the following mathematical assumptions: assuming query $Q$ and the key vector $K$ in SNNs are independent and share the same firing rate. Then we examine the $\frac{\|q\|}{\|k\|}$. Let $\boldsymbol{x} = (x_1, x_2, \ldots, x_D) \in \{0, 1\}^D$ represent $\boldsymbol{q}$ or $\boldsymbol{k}$, where each element $\boldsymbol{x}_i$ takes the value 1 with probability $p$. The square of the magnitude follows a binomial distribution:

$$\|\boldsymbol{x}\|^2 = \sum_{i=1}^{D} \boldsymbol{x}_i^2 = \sum_{i=1}^{D} \boldsymbol{x}_i \sim \mathrm{B}(D, P), \tag{18}$$

Its probability is given by:

$$P(\|\boldsymbol{x}\|^2 = k) = \binom{D}{k} p^k (1-p)^{D-k}, \quad k = 0, 1, 2, \ldots, D. \tag{19}$$

We randomly select a $\boldsymbol{q}$ and a $\boldsymbol{k}$ from this distribution and calculate their magnitude ratio $R = \frac{\|\boldsymbol{q}\|}{\|\boldsymbol{k}\|}$. Without considering the case when $\|\boldsymbol{k}\| = 0$, the calculation proceeds as follows:

$$\begin{aligned} P(R = r) &= P(R^2 = r^2) \\ &= \sum_{k=0}^{n} \sum_{l=1}^{n} \mathbf{1}\left(\frac{k}{l} = r^2\right) P(X = k) P(Y = l) \\ &= \sum_{k=0}^{n} \sum_{l=1}^{n} \mathbf{1}\left(\frac{k}{l} = r^2\right) \binom{n}{k}\binom{n}{l} p^{k+l}(1-p)^{2n-k-l} \end{aligned} \tag{20}$$

where $\mathbf{1}\left(\frac{k}{l} = r^2\right)$ is the indicator function, which equals 1 when $\frac{k}{l} = r^2$ and 0 otherwise.

Given the complexity of this distribution, we employ experimental simulation for approximation. Referencing the data shown in Fig. 1, we set $p = 0.15$ and $D = 128$. As the Dot-Product operation is symmetric, we adjust our calculation to ensure that the numerator is always greater than or equal to the denominator, enhancing the clarity of our visualization. Specifically, we compute $\frac{\|q\|}{\|k\|}$ when $\|\boldsymbol{q}\| > \|\boldsymbol{k}\|$, and $\frac{\|k\|}{\|q\|}$ otherwise. The simulation results are shown in Fig.6(b). Clearly, the distribution of $\frac{\|q\|}{\|k\|}$ is notably disordered. For comparison, we conduct the same assumptions and simulations in the self-attention module of ANN. Let $\boldsymbol{x} = (\boldsymbol{x}_1, \boldsymbol{x}_2, \ldots, \boldsymbol{x}_D) \in \mathbb{R}^D$ represent $\boldsymbol{q}$ or $\boldsymbol{k}$. Then its magnitude is given by $\|\boldsymbol{x}\| = \sqrt{\sum_{i=1}^{D} \boldsymbol{x}_i^2}$. Similarly, we randomly select a $\boldsymbol{q}$ and a $\boldsymbol{k}$ and simulate the distribution of their magnitude ratio $R'$. Based on the data in Fig. 1, we approximate each element $x_i$ as independently normally distributed with $\boldsymbol{x}_i \sim N(35, 10)$. The results are shown in Fig.6(c). By calculating the variance of $\frac{\|q\|}{\|k\|}$, it is found to be approximately 0.2322 in SNNs and only around 0.00844 in ANNs. This indicates significant magnitude fluctuations in SNNs, revealing a high degree of instability. As a result, the efficiency and effectiveness of the Dot-Product computation are negatively impacted.

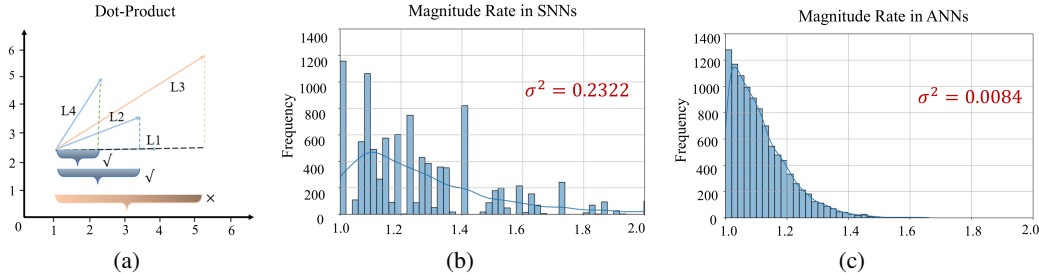

Figure 6: (a) The impact of varying magnitudes on the results of Dot-Products. (b) and (c) The distribution of magnitude entropy for $\frac{\|\boldsymbol{q}\|}{\|\boldsymbol{k}\|}$ in ANNs and SNNs.

## B  CROSS-ENTROPY FOR BETTER RELEVANCE COMPUTATION

We calculate the relevance in spatial dimensions separately for the $Q$ and $K$ vectors at different moments. The vectors q $\in Q$ and k $\in K$, and $p_q$ and $p_K$ represents the spike firing rate for them. Here, we introduce a cross-entropy method to more effectively compute the relevance between $q$ and $k$ vectors:

$$\mathcal{H}(q, k) = - \left[ p_\mathrm{q} \log p_\mathrm{k} + (1 - p_\mathrm{q}) \log (1 - p_\mathrm{k}) \right]. \tag{21}$$

The former term quantifies the degree of relevance when predictions are positive, while the latter reflects the relevance of negative predictions. When using cross-entropy as a measure of relevance, spike trains are first normalized and then transformed into probability distributions. Spike trains typically comprise only two states: spike and silence. Therefore, the probability distribution primarily reflects the spike firing rate. This measurement approach focuses on comparing the differences between two probability distributions, disregarding their magnitudes. In summary, cross-entropy is an effective distribution-based tool for assessing the relevance of spike trains, allowing for more precise comparisons and evaluations of similarities and differences between $Q$ and $K$.

**Approximation Methodology:** Although cross-entropy is an effective method for measuring relevance, a comprehensive analysis of spike states and silent periods may reduce the system's sensitivity. This is primarily because excessive focus on inactive silent periods can obscure critical information present during active spike periods when dealing with sparse spike trains. Given our focus on the spike states rather than silent periods in subsequent analyses, we can neglect the $(1 - p_\mathrm{q}) \log(1 - p_\mathrm{k})$ component. Therefore, $\mathcal{H}(q, k)$ can be simplified as follows:

$$\mathcal{H}(a, b) \approx -p_\mathrm{q} \log p_\mathrm{k}, \tag{22}$$

where $p_\mathrm{a}$ and $p_\mathrm{b}$ respectively represent the spike firing rates. However, the $\log(\cdot)$ function introduces non-linear operations that compromise the energy efficiency of SNNs. To address this, we propose a further approximation and simplification.

As described in the previous section, within the Transformer blocks of the SNNs, the spike firing rate of the $Q$ vector and $K$ vector primarily range from 10% to 20%. Consequently, we perform a Taylor expansion of $\log(x)$ at $x = 0.15$. This can be expressed as:

$$\log(x) \approx \log^{(0)}(0.15) + \ldots + \frac{\log^{(n)}(0.15)}{n!} \cdot (x - 0.15)^n \tag{23}$$

Here, $\log^{(n)}$ function denotes the result of the $n$-th order derivative of the $\log(\cdot)$ function. Given that $x$ is essentially between 0.1 and 0.2, The terms $(P_Q - 0.15)^2$ and higher-order terms are very small, which can be neglected. Consequently, $\mathcal{H}(A, B)$ can consider only the first term of the expansion:

$$\log(x) \approx \log^{(0)}(0.15) + \frac{\log^{(1)}(0.15)}{1!}(x - 0.15) \approx kx + b \tag{24}$$

In the training process of SNNs, since $k$ and $b$ can be learned as weights and biases, we use $x$ to replace $\log(x)$ to simplify computations. Although this may introduce slight errors, it avoids nonlinear operations and significantly enhances the network's energy efficiency.

## C  SACCADIC TEMPORAL INTERACTION

**Saccadic mechanism in biologic Vision:** Numerous neuroscience findings(Melcher & Morrone, 2003; Binda & Morrone, 2018; Guadron et al., 2022) confirm that the eyes do not acquire all details of a scene simultaneously. Instead, attention is focused on specific regions of interest (ROIs) through a series of rapid saccadic called saccades. Each saccade lasts for a very brief period—typically only tens of milliseconds—allowing the retina's high-resolution area to sequentially align with different visual targets. This dynamic saccadic mechanism enables the visual system to process information efficiently by avoiding redundant processing of the entire visual scene.

**Other similar works inspired by visual mechanisms**: Zhao et al. (2021) introduces a model utilizing a retina-inspired spiking camera to enhance image clarity in high-speed motion scenarios. McIntosh et al. (2016) explores how deep convolutional neural networks can model the retina's response to natural scenes. Tanaka et al. (2019) discusses the use of deep learning models to understand the computational mechanisms of the retina. These advanced features of biological vision effectively inform the rational design of deep neural networks, promoting the efficient integration of biological and machine intelligence.

**Leaky Integrate-and-Fire (LIF) neuron model**: In the LIF models, resetting and decay mechanisms significantly impair the neuron's ability to retain long-term historical information. The model's dynamics are described by the differential equation:

$$\tau_m \frac{\mathrm{d}V}{\mathrm{d}t} = -(V(t) - V_{\text{rest}}) + RI(t), \tag{25}$$

where $V(t)$ is the membrane potential, $V_{\text{rest}}$ is the resting potential, $\tau_m$ is the membrane time constant which influences decay rate, $R$ is the membrane resistance, and $I(t)$ is the input current. This equation illustrates how the membrane potential responds to input currents and decays towards $V_{rest}$. When the membrane potential $V(t)$ reaches the threshold $V_{th}$, the neuron fires and resets the potential to $V_{\text{reset}}$. This resetting process can be mathematically described as:

$$V(t^+) = V_{\text{reset}} \quad \text{if} \quad V(t) \geq V_{th}, \tag{26}$$

where $t^+$ is the time immediately following the spikes. This resetting not only disrupts the continuity of $V(t)$ but also eliminates all accumulated potential exceeding the threshold. Moreover, in the absence of input ($I(t) = 0$), the decay mechanism mercilessly forces the membrane potential to exponentially converge to the resting potential $V_{\text{rest}}$, following the equation:

$$V(t) = V_{\text{rest}} + (V_0 - V_{\text{rest}})e^{-\frac{t}{\tau_m}}, \tag{27}$$

where $V_0$ is the initial potential. This decay process gradually diminishes the stored information in the neuron, causing the accumulated potential to disappear quickly over time. It severely limits the neuron's ability to maintain historical information. To address this issue, we specifically designed saccadic spiking neurons without decay and reset mechanisms. The training and inference processes are described as follows.

## D  SACCADIC NEURONS

**Training Phase:** In the training process of SNN-ViT, the information from all timesteps is inputted in parallel. During this phase, the dynamics of the saccadic spiking neuron can be described as follows:

$$\begin{cases} \mathbf{H} = \mathbf{M}_w \mathbf{S} \\ \mathbf{S} = \Theta\left(\mathbf{H} - \mathbf{V}_{th}\right) \end{cases}, \quad \mathbf{M}_w \begin{pmatrix} w_{11} & \cdots & 0 \\ \vdots & \ddots & \vdots \\ w_{n1} & \cdots & w_{nn} \end{pmatrix}, \tag{28}$$

These neurons utilize a learnable lower triangular matrix $\mathbf{M}$ to integrate information across all timesteps without any loss of historical data, enhancing long-term memory and processing capabilities. As $\mathbf{M}$ is a lower triangular matrix, it naturally associates earlier inputs with current states, facilitating a comprehensive understanding of the whole visual scene over time. This method ensures that contributions from each timestep are precisely modulated and accumulated through matrix $\mathbf{M}$.

This approach not only prevents information loss due to decay and resets but also allows neurons to utilize all historical data more effectively for decision-making. Such capability markedly improves their capacity to understand the context of entire image scenes.

**Inference Phase:** During the training phase, all information is inputted into the network simultaneously, enabling efficient interaction through direct matrix multiplication. However, this parallel processing approach incurs significant resource expenditure, which is unfriendly to resource-constrained edge devices. Therefore, to maintain the energy efficiency and asynchronous processing advantages of SNNs, we propose an asynchronous decouple method that only computes the input at the current timestep. The dynamics of the inference process can be described as follows:

$$\begin{cases} H[t] = \mathcal{S}[t] \\ S[t] = \Theta\left(H[t] - \mathbf{M}_w^{-1}\mathbf{V}_{th}[t]\right) \end{cases} \tag{29}$$

During the inference process, the threshold $\mathbf{V}_{th}$ varies at each moment, thus $\mathbf{V}_{th} \in \mathbb{R}^T$. To ensure the existence of the inverse $\mathbf{M}_w^{-1}$ for the matrix $\mathbf{M}_w$, certain constraints must be imposed on $\mathbf{M}_w$. It can be described as follows:

$$\det(\mathbf{M}_w) = m_{11} \times m_{22} \times \ldots \times m_{nn} \neq 0 \tag{30}$$

By incorporating the inverse of M into the threshold of the saccadic spiking neurons, we ensure temporal decoupling between H and S. Specifically, $\mathbf{H}[t]$ only requires input from $\mathbf{S}[t]$, facilitating asynchronous inference. Additionally, dynamic thresholds at each moment enrich the dynamical properties of the spiking neurons. This approach effectively highlights the spatio-temporal attributes of SNNs and ensures efficient performance of vision tasks on resource-constrained edge devices.

## E  ABLATION STUDIES ON SSSA

We add ablation studies on the two key components of the SSSA module: (1) Replacing Distribution-Based Spatial Similarity Computation with traditional Dot Product (DP) similarity; (2) Replacing saccadic neurons with LIF neurons. Finally, we also compare the performance of versions V1 and V2. Experiments are performed on the CIFAR100 dataset, and the results are presented in the following Table.5.

Table 5: Ablation Studies on SSSA.

| Model | Param (M) | Complexity | Acc (%) |
|---|---|---|---|
| SSA | 5.76M | $\mathcal{O}(N^2D)$ | 76.95 |
| SSSA+DP | 5.52M | $\mathcal{O}(N^2D)$ | 77.12 |
| SSSA+LIF | 5.52M | $\mathcal{O}(D)$ | 78.84 |
| SSSA-V1 | 5.52M | $\mathcal{O}(N^2)$ | 79.71 |
| SSSA-V2 | 5.52M | $\mathcal{O}(D)$ | 79.60 |

The SSSA+DP shows almost no performance improvement compared to the baseline (Zhou et al., 2023b). This outcome underscores that effective spatial similarity computation is the foundation for subsequent saccadic interactions. Then, substituting saccadic neurons with LIF neurons led to an approximate 0.8% decrease in performance relative to SSSA. This demonstrates that saccadic interactions can indeed enhance performance. Finally, while there is virtually no performance disparity between V1 and V2, the computational complexity of V2 is only $\mathcal{O}(D)$. In summary, our SSSA-V2 module achieves an optimal trade-off between computational complexity and performance.

## F  EXPERIMENT SETTING FOR IMAGE CLASSIFICATION

On the ImageNet-1K classification benchmark, we propose an architecture according to Spike-driven V2 (Yao et al., 2024a). As illustrated in Table 6, our model introduces the GL-SPS part and SSSA block, which respectively replace the Patch Embedding block and self-attention computation blocks. The architecture of SNN-ViTs primarily consists of four stages. Each stage includes

GL-SPS encoding operations and SSSA module, facilitating efficient and precise visual information processing. Specifically, in the initial stage, the input sequence $I \in \mathbb{R}^{3 \times H \times W}$ is processed through a GL-SPS layer, which encodes it into $X \in \mathbb{R}^{C \times \frac{H}{2} \times \frac{W}{2}}$. Then, the encoded images are input into a spiking saccadic self-attention block to enhance feature extraction. This block comprises a SSSA module and an MLP layer in the channel dimension. Moreover, the output will be input to the GL-SPS layer of the next stage which has a similar operation to the previous stage. Additionally, residual connections are applied to membrane potentials in both SSSA module and MLP layer. Finally, the model is processed through a fully connected layer (FCH) to obtain the final classification output.

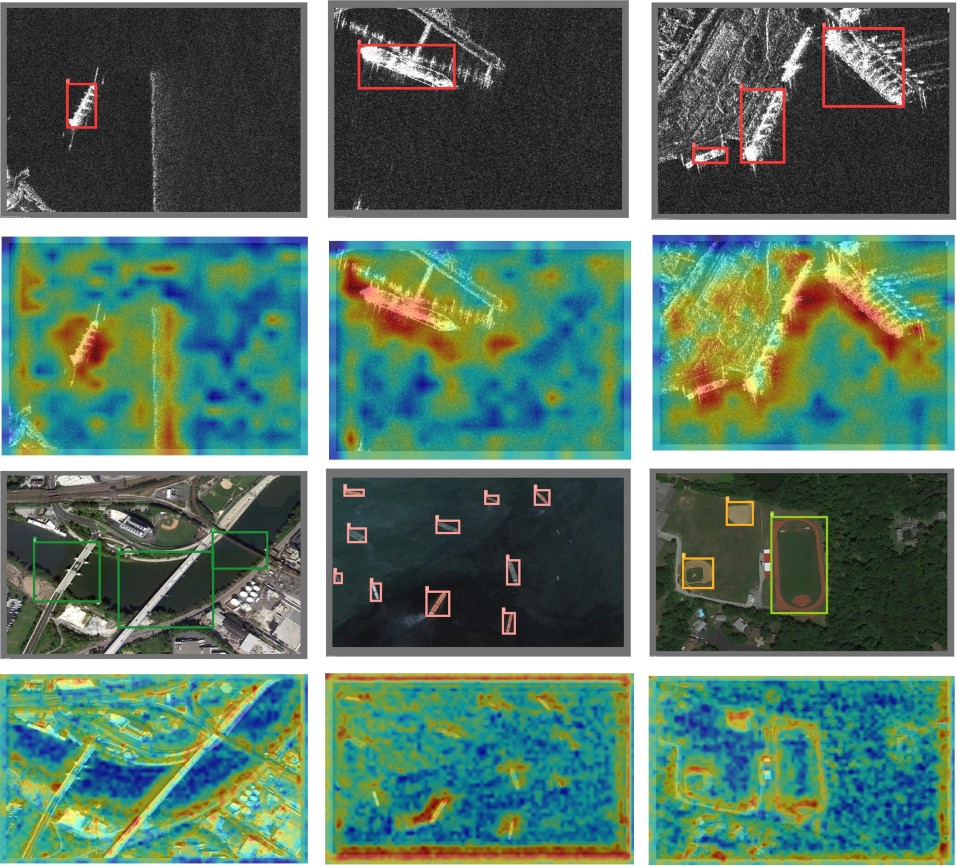

Figure 7: The detection performance and heatmaps of SNN-ViT-YOLO on the SSDD and NWPU VHR-10 datasets.

## G  EXPERIMENT DETAILS FOR REMOTE TARGET DETECTION

To further validate the capabilities of SNN-ViT across diverse task environments, this study conducts extensive experiments in remote sensing object detection. Specifically, the SNN-ViT is integrated into the widely utilized YOLO-v3 detection framework. Experiments are carried out on a high-performance computing platform equipped with an NVIDIA RTX4090 GPU, using Stochastic Gradient Descent (SGD) as the optimization algorithm. The initial learning rate is set at $1 \times 10^{-2}$, adjusted according to a polynomial decay strategy. The entire training process spans 300 epochs on the NWPU-VHR-10 and SSDD datasets, ensuring comprehensive learning and adaptation to data characteristics.

As illustrated in Fig.7, the SNN-ViT-YOLO model exhibits significant performance advantages on the NWPU test set, effectively pinpointing critical target points. This confirms its viability and efficiency in practical applications. The specific configurations of the network are meticulously detailed

Table 6: The details of experiment setting for ImageNet-1K

| Stage | # Tokens | Layer Specification | | | 14M | 30M | 53M |
|---|---|---|---|---|---|---|---|
| 1 | $\frac{H}{2} \times \frac{W}{2}$ | Downsampling | GL-SPS | Conv | $7 \times 7$ stride 2 | | |
| | | | | DConv | $3 \times 3$ stride 2 dilation 2 | | |
| | | | | Dim | 32 | 48 | 64 |
| | | Attention-based SNN block | SSSA | Conv | $3 \times 3$ stride 1 | | |
| | | | Channel Conv | Conv | $1 \times 1$ stride 1 | | |
| | | | | Conv ratio | 4 | | |
| | $\frac{H}{2} \times \frac{W}{2}$ | Downsampling | GL-SPS | Conv | $3 \times 3$ stride 1 | | |
| | | | | DConv | $3 \times 3$ stride 2 dilation 2 | | |
| | | | | Dim | 64 | 96 | 128 |
| | | Attention-based SNN block | SSSA | Conv | $3 \times 3$ stride 1 | | |
| | | | Channel Conv | Conv | $1 \times 1$ stride 1 | | |
| | | | | Conv ratio | 4 | | |
| 2 | $\frac{H}{4} \times \frac{W}{4}$ | Downsampling | GL-SPS | Conv | $3 \times 3$ stride 2 | | |
| | | | | DConv | $3 \times 3$ stride 2 dilation 2 | | |
| | | | | Dim | 128 | 192 | 256 |
| | | Attention-based SNN block | SSSA | Conv | $3 \times 3$ stride 1 | | |
| | | | Channel Conv | Conv | $1 \times 1$ stride 1 | | |
| | | | | Conv ratio | 4 | | |
| 3 | $\frac{H}{8} \times \frac{W}{8}$ | Downsampling | GL-SPS | Conv | $3 \times 3$ stride 2 | | |
| | | | | DConv | $3 \times 3$ stride 2 dilation 2 | | |
| | | | | Dim | 256 | 384 | 512 |
| | | Attention-based SNN block | SSSA | Conv | $3 \times 3$ stride 1 | | |
| | | | Channel Conv | Conv | $1 \times 1$ stride 1 | | |
| | | | | Conv ratio | 4 | | |
| 4 | $\frac{H}{16} \times \frac{W}{16}$ | Downsampling | GL-SPS | Conv | $3 \times 3$ stride 2 | | |
| | | | | DConv | $3 \times 3$ stride 2 dilation 2 | | |
| | | | | Dim | 256 | 384 | 512 |
| | | Attention-based SNN block | SSSA | Conv | $3 \times 3$ stride 1 | | |
| | | | Channel Conv | Conv | $1 \times 1$ stride 1 | | |
| | | | | Conv ratio | 4 | | |

in Table G. In this configuration, the expansion ratio of the Multi-Layer Perceptron (MLP) is fixed at 4 to achieve an optimal balance between computational efficiency and model performance. This network architecture involves feeding the $P_3$, $P_4$, and $P_5$ feature maps—derived from intermediate layers of the network—into the detection heads of the YOLO-v3 framework.

Table 7: Configurations of SNN-ViT on object detection.

| Stage | # Tokens | Layer Specification | | | Model |
|---|---|---|---|---|---|
| P1 | $\frac{H}{2} \times \frac{W}{2}$ | Downsampling | GL-SPS | Conv | $3 \times 3$ stride 2 dilation 2 |
| | | | | DConv | $7 \times 7$ stride 2 |
| | | | | Dim | 64 |
| | | Conv-based SNN block | Conv Layer | Conv | $3 \times 3$ stride 2 dilation 1 |
| | | | | Dim | 64 |
| | | | Channnel Conv | Conv | $1 \times 1$ stride 1 |
| | | | | Conv ratio | 4 |
| P2 | $\frac{H}{4} \times \frac{W}{4}$ | Downsampling | GL-SPS | Conv | $3 \times 3$ stride 2 dilation 2 |
| | | | | DConv | $7 \times 7$ stride 2 |
| | | | | Dim | 128 |
| | | Conv-based SNN block | Conv Layer | Conv | $3 \times 3$ stride 2 dilation 1 |
| | | | | Dim | 128 |
| | | | Channnel Conv | Conv | $1 \times 1$ stride 1 |
| | | | | Conv ratio | 4 |
| P3 | $\frac{H}{8} \times \frac{W}{8}$ | Downsampling | GL-SPS | Conv | $3 \times 3$ stride 2 dilation 2 |
| | | | | DConv | $7 \times 7$ stride 2 |
| | | | | Dim | 256 |
| | | Attention-based SNN block | Conv Layer | Conv | $3 \times 3$ stride 2 dilation 1 |
| | | | | Dim | 256 |
| | | | Channnel Conv | Conv | $1 \times 1$ stride 1 |
| | | | | Conv ratio | 4 |
| P4 | $\frac{H}{16} \times \frac{W}{16}$ | Downsampling | GL-SPS | Conv | $3 \times 3$ stride 2 dilation 2 |
| | | | | DConv | $7 \times 7$ stride 2 |
| | | | | Dim | 256 |
| | | Attention-based SNN block | Conv Layer | Conv | $3 \times 3$ stride 2 dilation 1 |
| | | | | Dim | 256 |
| | | | Channnel Conv | Conv | $1 \times 1$ stride 1 |
| | | | | Conv ratio | 4 |
| P5 | $\frac{H}{32} \times \frac{W}{32}$ | Downsampling | GL-SPS | Conv | $3 \times 3$ stride 2 dilation 2 |
| | | | | DConv | $7 \times 7$ stride 2 |
| | | | | Dim | 512 |
| | | Attention-based SNN block | Conv Layer | Conv | $3 \times 3$ stride 2 dilation 1 |
| | | | | Dim | 512 |
| | | | Channnel Conv | Conv | $1 \times 1$ stride 1 |
| | | | | Conv ratio | 4 |

