# OpenReview forum: "Spiking Vision Transformer with Saccadic Attention"
_ICLR.cc/2025/Conference — ICLR 2025 Poster_

### Official Review · Reviewer_pPD7 · 2024-10-30

**Soundness:** 3
**Presentation:** 3
**Contribution:** 4
**Rating:** 8
**Confidence:** 5

**Summary:**

This paper introduces a biologically inspired (Spike-based ViTs) inspired by biological visual saccadic attention mechanisms. The study thoroughly investigates two challenges for current SNN-based Transformer: degraded correlation computation and limited temporal dimensionality. To address these issues, the authors propose a distribution-based method for correlation computation and a saccadic time interaction strategy. Notably, the introduced SSSA module has O(d) computational complexity.

**Strengths:**

1. Biological Inspiration: The model is well-motivated by biological mechanisms, which is a strong point given the effectiveness of biological systems in visual tasks.
2. Low Computational Complexity: The SSSA method achieves exceptional performance while maintaining linear computational complexity.
3. Correlation Degradation Analysis: The discussion on the degradation of correlation computation with binary spike trains is clear and comprehensible, with theoretical and experimental validations.

**Weaknesses:**

1. Expression Details: Although the paper is generally clear, some symbols in the equations, like the QK subscripts in Equation 1 and the bolding of H in Equation 8, should be simplified for better distinctions
2. Energy Consumption Analysis: The energy consumption in the table.3 require further clarification.
3. Experimental Validation: Results should be verified through multiple trials, including error bars to enhance data reliability.

**Questions:**

1. I am curious about the time cost associated with the training phase. Given that SNNs are known for their substantial training overhead. Detailed information on the resource requirements and approximate time costs during network training should be discussed in the appendix.
2. I am very interested in the "Saccadic Neurons." Could the authors explain how the saccadic attention mechanism is implemented specifically? Additionally, I would appreciate an example detailing the processes of parallel training and serial decoupling.
3. Could the authors explain if the code implementations for SSSA-V1 and SSSA-V2 are identical, given their mathematical equivalence? Which method performs better?

---

> ### Author Response · Authors · 2024-11-23
>
> Thank you for your insightful feedback. We address each of your points in turn.
> ## Weakness 1: Expression details.
>
> Thank you for your suggestions. We clarify Equation 1 as follows:
> $$\text{Dot-Product}\( \mathcal{Q}_i, \mathcal{K}_i \) = \sum\_{j=1}^{D} \mathcal{Q}\_{ij} \mathcal{K}\_{ij}$$
> $D$ is the dimension of both vectors, $\mathcal{Q}\_{ij}$ and $\mathcal{K}\_{ij}$ refer to the  $j$-th elements of these vectors, respectively. Additionally,  the bolded letters $\textbf{H}$ and $\textbf{V}\_{th}$ represent the membrane potential and threshold that include temporal dimensions, whereas the non-bold letters denote $H[t]$ and $V\_{th}[t]$ at a specific timestep.
>
> ## Weakness 2: Further energy consumption analysis.
> Thank you for your feedback. Compared to SSA, our SSSA exhibits lower computational complexity and consequently fewer SOP operations. The Table below presents a comparative analysis of energy consumption across different SSA configurations under the conditions of Head=8, D=512, N=196, and T=4. Notably, our SSSA demonstrates reduced SOPs and lower computational energy consumption. However, the overall energy consumption of the SNN-VIT varies due to differences in network performance and architecture, rendering the comparative results less pronounced.
>
> |   Architecture   | Spike self-attention |       Complexity       | SOP in Attention | Energy (uJ) | Acc. (%) |
> | :--------------: | :------------------: | :--------------------: | :--------------: | :---------: | :------: |
> | Spikformer-8-512 |         SSA          |  $\mathcal{O}(N^2D)$   |      39.34M      |    35.04    |  74.81   |
> | STSAformer-8-512 |         STSA         | $\mathcal{O}(T^2N^2D)$ |     178.68M      |   160.81    |  74.79   |
> | Meta-SpikeFormer |        SDSA-4        |  $\mathcal{O}(ND^2)$   |      12.58M      |    11.32    |  79.70   |
> |       Ours       |         SSSA         |    $\mathcal{O}(D)$    |      1.21M       |    1.089    |  80.23   |
>
> ## Weekness 3: Experimental Validation.
> Thank you for your suggestions. While we conducted multiple experiments, we inadvertently did not calculate the error bars. As shown in the table below, all experiments are conducted at least five times to ensure reliability.
>
> |   Dataset   | Param.(M) |       Acc.(%)        |
> | :---------: | :-------: | :------------------: |
> |    |   14.28   | $74.66 \pm 0.25$ |
> |      ImageNet        |   20.83   | $76.87 \pm 0.65$ |
> |             |   35.75   | $80.23 \pm 0.32$ |
> |   CIFAR10   |   6.42    | $96.10 \pm 0.25$ |
> |  CIFAR100   |   6.42    | $80.10 \pm 0.35$ |
> | CIFAR10DVS  |   1.52    | $82.30 \pm 0.15$ |
> | NWPU VHR-10 |     76.20     | $89.40 \pm 0.15$ |
> |    SSDD     |     76.20     | $97.00 \pm 0.24$ |

---

> ### Author Response · Authors · 2024-11-23
>
> ## Question 1:
> Thank you for your suggestions. We have compiled the resource requirements and training durations for the SNN-ViT across various tasks, detailed as follows:
>
> - For CIFAR-10 and DVS-CIFAR10, training was conducted using a single NVIDIA RTX 4090 GPU, with each task requiring approximately 6-8 hours.
> - For CIFAR-100, we utilized a four-GPU setup with NVIDIA RTX 4090s, also completing in about 6-8 hours.
> - For ImageNet1K, the training was performed using four NVIDIA A100 GPUs , taking approximately 5-7 days.
> ## Question 2: The details of training and inference for saccadic neurons.
> We apologize if Eq.8 causes any confusion in reading this manuscript. Here, we will separately introduce the training and inference processes for the saccadic neurons.
> **A:** We apologize for any confusion caused by our wording. The bolded letters $\textbf{H}$ and $\textbf{V}\_{th}$ represent the membrane potential and threshold that include temporal dimensions, whereas the non-bold letters denote $H[t]$ and $V_{th}[t]$ at a specific timestep. Additionally, to more clearly articulate the training and inference processes of our saccadic neurons, we provide separate explanations for each:
>
> **training process:**
> The dynamic of saccadic neurons can be described as follows:
> $$
> \left\\{
> \begin{array}{l}
> \textbf{H} =  \textbf{M}_w \mathcal{P}atch, \\\\
> \textbf{S} =  \left\\{
> \begin{array}{l}
> 1, & \text{if }\textbf{H} \geq \textbf{V}\_{th}, \\\\
> 0, & otherwise.
> \end{array}
> \right.
> \end{array}
> \right.
> $$
>
> $\mathcal{P}atch \in \mathbb{R}^{T \times N}$ represents the spatial importance of different regions, $\textbf{M}\_w$ is a lower triangular matrix, which can represent $\textbf{M}\_w = \left\(
> \begin{matrix}
> w_{11}&\cdots&0\\\\
> \vdots&\ddots&\vdots\\\\
> w_{n1}&\cdots&w_{nn}\\\\
> \end{matrix}
> \right\)$. $\textbf{S}$ are spike trains, generated by performing an element-wise comparison between the $\textbf{H}$ and $\textbf{V}_{th}$.
>
> **Inference Process:**
>
> **Lemma:** A lower triangular matrix is invertible if and only if all its diagonal elements are nonzero.
>
> The diagonal elements of $\textbf{M}\_w$ represent the proportional weights of $\mathcal{P}atch$ at the current time step, and we ensure they are nonzero during training. Therefore, $\textbf{det}\left( \textbf{M}\_{w} \right) \neq 0$, then $\textbf{M}\_w$​ is Invertible.
> $$
> \left\\{
> \begin{array}{l}
> \textbf{H}[t] =   \mathcal{P}atch[t], \\\\
> \textbf{S} =  \left\\{
> \begin{array}{l}
> 1, & \text{if }\textbf{H}[t] \geq \textbf{M}_w^{-1}\textbf{V}'\_{th}[t], \\\\
> 0, & otherwise.
> \end{array}
> \right.
> \end{array}
> \right.
> $$
>
> where $\textbf{V}'\_{th}=\textbf{M}_{w}^{-1} \textbf{V}\_{th}$ represents the threshold $\textbf{V}\_{th}$ at different timesteps. Through this approach,  we implement a sequential inference process and ensure the equivalence of the training and inference processes.
>
> ## Question 3:  Which method performs better? SSSA-V1 or SSSA-V2 .
> Our subsequent experiments are conducted on SSSA-V2.  In the training process, SSSA-V2 and V1 exhibit a linear scaling relationship with virtually no performance loss. The distinction between them emerges during the inference process, where V2 integrates $(\mathcal{K}^T \times L)$ into the threshold $V_{th}$ of the saccadic neurons as a constant value, whereas in V1, it is data-driven. We have also conducted experiments that show minimal performance disparity between the two versions and have successfully reduced the computational complexity from $\mathcal{O}\left(N^2\right)$to $\mathcal{O}\left(D\right)$.
>
>
> |  model   | Param (M) |           Complexity            |   Acc (%)     |
> | :------: | :-------: | :-----------------------------: | :-----------: |
> | SSSA-V1  | **5.52M** |  $\mathcal{O}\left(N^2\right)$  |     79.71     |
> | SSSA-V2  | **5.52M** |   $\mathcal{O}\left(D\right)$   |     79.60     |

---

### Official Review · Reviewer_y4bx · 2024-11-01

**Soundness:** 3
**Presentation:** 3
**Contribution:** 2
**Rating:** 6
**Confidence:** 4

**Summary:**

This paper analyzes the spatial relevance and temporal interaction in vanilla SNN-based ViTs, indicating that the mismatch between vanilla self-attention mechanisms and spatio-temporal spike trains may be one of the reasons for the performance gap between SNN-based and ANN-based ViTs. To address these issues, this paper proposes a biologically inspired SSSA mechanism. In both the spatial and temporal domains, SSSA demonstrates a stronger capability for information processing. Furthermore, the SNN-ViTs built upon SSSA also achieve SOTA performance with linear computational complexity, highlighting the energy efficiency potential of SNNs in practical applications.

**Strengths:**

1. This paper offers a plausible explanation for the performance gap between SNN-based and ANN-based ViTs and proposes an efficient mechanism to replace the vanilla self-attention mechanism. SNN-ViTs built on the proposed method achieves both high-performance and energy-efficience.

2. The temporal interaction within SSSA addresses a significant question: What are the benefits of utilizing SNNs with temporal structures for processing image data that lacks inherent temporal structures?

**Weaknesses:**

1. Prior to their combination with transformer architecture, the performance gap between SNNs and ANNs already existed, suggesting that an enhanced attention mechanism alone cannot fully mitigate the performance decrement inherent to spiking neuron. Therefore, Table 1-3 ought to include the performances of SOTA ANN-based ViTs, to clearly illustrate the performance gap between SNN-based and ANN-based models. Although this reality may be discouraging, a precise comprehension of this gap is conducive to the practical implementation of SNN research.

2. The introduction to biological saccadic mechanisms is inadequate, especially due to the absence of a mathematical formulation. This results in a seemingly tenuous connection between the methodology and its biological inspiration.

**Questions:**

1. It is required to include the performances of SOTA ANN-based ViTs in Tables 1-3 for a direct comparison with the SNN-based ViTs results.

2. The introduction of biological saccadic mechanisms and the rationale behind the inspiration can be further elaborated. Specifically, this should include the research history of biological saccadic mechanisms, an overview of related works on algorithms inspired by other visual mechanisms, and concrete examples illustrating how the mathematical formulation of the proposed method correlates with the characteristics of biological saccades.

---

> ### Author Response · Authors · 2024-11-23
>
> We greatly appreciate your recognition of the innovative aspects and motivation of our work. We will respond to each of your suggestions in turn.
> ## Weakness & Questions 1: Include the performances of SOTA ANN-based ViTs.
>
> **A**: Thank you for your valuable suggestions. We will revise the manuscript to include comprehensive performance comparisons with state-of-the-art ANN-based ViTs for Table 1-3. This addition allows for a more rigorous evaluation of our method's effectiveness relative to existing approaches. For example, the details of comparisons for ImageNet-1K are shown as follows：(Table 1-3)
>
> |       Model       | Type  | Timesteps | Param (M) |           Complexity            | Accuracy (%) |
> | :---------------: | :---: | :-------: | :-------: | :-----------------------------: | :----------: |
> |    ViT-B/16[1]    | ANN   |     1     |    86M    | $\mathcal{O}\left(N^2 D\right)$ |     77.9     |
> |    ViT-L/16[1]    | ANN   |     1     |   307M    | $\mathcal{O}\left(N^2 D\right)$ |     76.5     |
> |     Swin-T[2]     | ANN   |     1     |    29M    | $\mathcal{O}\left(N^2 D\right)$ |     81.3     |
> |     Swin-S[2]     | ANN   |     1     |    50M    | $\mathcal{O}\left(N^2 D\right)$ |     83.0     |
> |     Swin-B[2]     | ANN   |     1     |    88M    | $\mathcal{O}\left(N^2 D\right)$ |     76.5     |
> | FLatten-Swin-T[3] | ANN   |     1     |    29M    |  $\mathcal{O}\left(ND^2\right)$ |     82.1     |
> | FLatten-Swin-S[3] | ANN   |     1     |    51M    |  $\mathcal{O}\left(ND^2\right)$ |     83.5     |
> | FLatten-Swin-B[3] | ANN   |     1     |    89M    |  $\mathcal{O}\left(ND^2\right)$ |     83.8     |
> |    SNN-ViT-256    | SNN   |     4     |   13.7M   |   $\mathcal{O}\left(D\right)$   |    74.66     |
> |    SNN-ViT-384    | SNN   |     4     |   30.4M   |     $\mathcal{O}\left(D\right)$ |    76.87     |
> |    SNN-ViT-512    | SNN   |     4     |   53.7M   |     $\mathcal{O}\left(D\right)$ |    80.23     |
>
> Reference：
>
> [1] Dosovitskiy, Alexey. "An image is worth 16x16 words: Transformers for image recognition at scale." arXiv preprint arXiv:2010.11929 (2020).
>
> [2] Liu, Z., Lin, Y., Cao, Y., Hu, H., Wei, Y., Zhang, Z., Guo, B. Swin transformer: Hierarchical vision transformer using shifted windows. In Proceedings of the IEEE/CVF international conference on computer vision (2021).
>
> [3] Han D, Pan X, Han Y, et al. Flatten transformer: Vision transformer using focused linear attention. Proceedings of the IEEE/CVF international conference on computer vision (2023).

---

> ### Author Response · Authors · 2024-11-23
>
> ## Weakness & Questions 2: introduction of biological saccadic mechanisms.
>
> **A**: Thank you for your suggestions. We add an explanation of biological saccadic mechanisms from three perspectives in the revised manuscript. (Appendix.C）
>
> 1.**Details of biological saccadic mechanisms**: Numerous neuroscience findings [1-3] confirm that the eyes do not acquire all the details of a scene simultaneously. Instead, attention is focused on specific regions of interest (ROIs) through a series of rapid saccadic called saccades. Each saccade lasts for a very brief period—typically only tens of milliseconds—allowing the retina's high-resolution area to align with different visual targets sequentially. This dynamic saccadic mechanism enables the visual system to process information efficiently by avoiding redundant processing of the entire visual scene.
>
> 2.**Other similar works inspired by visual mechanisms**: Zhao, Jing, et al.[4] introduce a model utilizing a retina-inspired spiking camera to enhance image clarity in high-speed motion scenarios. McIntosh, Lane, et al.[5] explore how deep convolutional neural networks can model the retina's response to natural scenes. Tanaka, Hidenori, et al.[6] discuss the use of deep learning models to understand the computational mechanisms of the retina. These advanced features of biological vision effectively inform the rational design of deep neural networks, promoting the efficient integration of biological and machine intelligence.
>
> 3.**Specific examples of saccadic interaction**:
>
> For the saccadic mechanism, two key processes are briefly included: (1). It focuses on a small part of the scene at each moment while ignoring other information. (2). The eye achieves a global understanding of the entire visual scene by continuously moving its focus at different moments, combined with a history of previous focal points.
>
> In our saccadic neurons, the salient patch selection process primarily aims to facilitate the first process by assessing the overall relevance of different patches to each other. It can be described as:
>
> $$\mathcal{P}atch = \sum_{j=1}^{n} \mathrm{CroAtt}\left(Q, K\right), \textbf{       }  \mathrm{CroAtt}\left(Q, K\right) \in \mathbb{R}^{T \times N \times N},$$
>
>  Higher-scoring patches indicate more critical spatial positions while lower-scoring ones are overlooked. This scoring provides the basis for choosing the patch at each moment.
>
> Subsequently, saccadic neurons decide which visual regions to select based on the current patch score and historical attention, achieving the second process.
> $$
> \text{Training}
> \begin{cases}
> \mathbf{H} = \mathbf{M}_w \mathcal{P}atch, \\\\
> \mathbf{S} = \Theta (\mathbf{H} - \mathbf{V}\_{th}),
> \end{cases} \quad
> \text{Inference}
> \begin{cases}
> \mathbf{H}[t] = \mathcal{P}atch [t] \\\\
> \mathbf{S}[t] = \Theta \left(\mathbf{H}[t] - \mathbf{M}_w^{-1}{V}\_{th}[t]\right)
> \end{cases}
> $$
> Notably, our saccadic neurons are specifically designed to integrate historical information into different thresholds at each moment, ensuring performance under full spike-driven conditions.
>
> Reference:
>
> [1] Melcher, David, and M. Concetta Morrone. "Spatiotopic temporal integration of visual motion across saccadic eye movements." Nature Neuroscience 6.8 (2003): 877-881.
>
> [2] Binda, Paola, and Maria Concetta Morrone. "Vision during saccadic eye movements." Annual review of vision science 4.1 (2018): 193-213.
>
> [3] Guadron, Leslie, A. John van Opstal, and Jeroen Goossens. "Speed-accuracy tradeoffs influence the main sequence of saccadic eye movements." Scientific Reports 12.1 (2022): 5262.
>
> [4] Zhao, J., Xiong, R., Xie, J., Shi, B., Yu, Z., Gao, W., & Huang, T. Reconstructing clear image for high-speed motion scene with a retina-inspired spike camera. IEEE Transactions on Computational Imaging, 8, 12-27, (2021).
>
> [5] McIntosh, L., Maheswaranathan, N., Nayebi, A., Ganguli, S., & Baccus, S. Deep learning models of the retinal response to natural scenes. Advances in neural information processing systems, 29, (2016).
>
> [6] Tanaka, H., Nayebi, A., Maheswaranathan, N., McIntosh, L., Baccus, S., & Ganguli, S. From deep learning to mechanistic understanding in neuroscience: the structure of retinal prediction. Advances in neural information processing systems, 32, (2019).

---

### Official Review · Reviewer_WEvM · 2024-11-02

**Soundness:** 3
**Presentation:** 3
**Contribution:** 3
**Rating:** 8
**Confidence:** 5

**Summary:**

This work draws on biological saccadic attention mechanisms to design a ViT with reduced computational overhead. It proposes a novel self-attention paradigm tailored for spikes, achieving a computational complexity of O(D). Extensive experimental results validate the model's advanced accuracy and general applicability.

**Strengths:**

Overall, this is an interesting work. The strengths are outlined as follows:
1. The approach is thoroughly presented, including detailed explanations of the design and operational logic of all modules. The method section is also specifically tailored to address the issues raised.
2. The analysis of the correlation computation among spike sequences is good, with solid theoretical backing and substantial experimental support.
3. The complexity of the proposed SSSA is O(D), which is commendable for reducing computational complexity and further leveraging the energy efficiency of SNNs.

**Weaknesses:**

1.Unclear Expression: I suggest author to provide a more detailed description of the workings of the Saccadic neurons in the main text. For example, in Equation (8), what is the difference between the bolded Vth​ and the non-bolded version? Additionally, in lines 302-308, why does asynchronous inference not require dependence on historical membrane potentials? I believe this is crucial for achieving lower computational complexity in SSSA.
2.The experiment details of SNN-ViT with YOLO-v3 in remote sensing target detection tasks is not clearly articulated. For example: How does the backbone network integrate with the classification head? What are the training environment and hardware requirements? These aspects should be further clarified in the appendix.
3.While the appendix is informative, it does not fully bridge the knowledge gap for those outside the specialty. I suggest adding detailed explanations for Appendices A and B.   Appendix A could be expanded to include why dot products in ANNs effectively measure Q-K correlation, along with their mathematical proof. Secondly, cross-entropy is an important method for measuring similarity in Appendix B—why is it not used in ANNs for this purpose, while dot products are preferred instead?

**Questions:**

1. I'm interested in the author's distribution-based approach to correlation calculation. Does this method assess the similarity between Q and K based on spike firing rates？ does it ensure that non-spike computations are not introduced?
2. Were all subsequent experiments conducted on SSSA-V2? Would V1 perform better when floating-point computations are involved?

---

> ### Author Response · Authors · 2024-11-23
>
> Thank you for your insightful feedback. We will address each of your points in turn.
> ## Weakness 1:  Unclear Expression.
> **A:** We apologize for any confusion caused by our wording. The bolded letters $\textbf{H}$ and $\textbf{V}\_{th}$ represent the membrane potential and threshold that include temporal dimensions, whereas the non-bold letters denote $H[t]$ and $V_{th}[t]$ at a specific timestep. Additionally, to more clearly articulate the training and inference processes of our saccadic neurons, we provide separate explanations for each:
>
> **training process:**
> The dynamic of saccadic neurons can be described as follows:
> $$
> \left\\{
> \begin{array}{l}
> \textbf{H} =  \textbf{M}_w \mathcal{P}atch, \\\\
> \textbf{S} =  \left\\{
> \begin{array}{l}
> 1, & \text{if }\textbf{H} \geq \textbf{V}\_{th}, \\\\
> 0, & otherwise.
> \end{array}
> \right.
> \end{array}
> \right.
> $$
>
> $\mathcal{P}atch \in \mathbb{R}^{T \times N}$ represents the spatial importance of different regions, $\textbf{M}\_w$ is a lower triangular matrix, which can represent $\textbf{M}\_w = \left\(
> \begin{matrix}
> w_{11}&\cdots&0\\\\
> \vdots&\ddots&\vdots\\\\
> w_{n1}&\cdots&w_{nn}\\\\
> \end{matrix}
> \right\)$. $\textbf{S}$ are spike trains, generated by performing an element-wise comparison between the $\textbf{H}$ and $\textbf{V}_{th}$.
>
> **Inference Process:**
>
> **Lemma:** A lower triangular matrix is invertible if and only if all its diagonal elements are nonzero.
>
> The diagonal elements of $\textbf{M}\_w$ represent the proportional weights of $\mathcal{P}atch$ at the current time step, and we ensure they are nonzero during training. Therefore, $\textbf{det}\left( \textbf{M}\_{w} \right) \neq 0$, then $\textbf{M}\_w$​ is Invertible.
> $$
> \left\\{
> \begin{array}{l}
> \textbf{H}[t] =   \mathcal{P}atch[t], \\\\
> \textbf{S} =  \left\\{
> \begin{array}{l}
> 1, & \text{if }\textbf{H}[t] \geq \textbf{M}_w^{-1}\textbf{V}'\_{th}[t], \\\\
> 0, & otherwise.
> \end{array}
> \right.
> \end{array}
> \right.
> $$
>
> where $\textbf{V}'\_{th}=\textbf{M}_{w}^{-1} \textbf{V}\_{th}$ represents the threshold $\textbf{V}\_{th}$ at different timesteps. Through this approach,  we implement a sequential inference process and ensure the equivalence of the training and inference processes.
>
> ## Weakness 2:  The experiment details of SNN-ViT with YOLO-v3.
> **A:** Feature maps from the backbone network are fed into detection heads through concatenation for multi-scale object detection. This hierarchical feature fusion strategy enables the model to capture objects at different scales and spatial contexts.
> The experimental framework was implemented using the PyTorch deep learning library. All experiments are conducted on a high-performance computing platform equipped with an NVIDIA RTX4090 GPU. This hardware configuration provided sufficient computational capacity for training our proposed architecture through all experimental phases.
>
> ## weekness 3: Further explanations for Appendix.
> **A:** (1) Why is the dot product used to compute similarity in attention mechanisms in ANNs?
>
> The dot-product is widely used in ANNs for its simplicity and effectiveness in capturing vector similarity. when vectors are normalized, it becomes cosine similarity, reflecting their angular relationship. Moreover, it is computationally efficient, leveraging matrix multiplication to handle large-scale data, ensuring high performance during training and inference.
>
> (2) Why is cross-entropy not used to compute similarity in attention mechanisms in ANNs?
>
> Cross-entropy measures the difference between probability distributions, but ANN vectors are not probability distributions—they may include negative values or elements that do not sum to one. Converting them using Softmax operation adds computational cost and distorts the original vector properties, reducing accuracy. In contrast, SNNs use binary data (0 or 1), where firing rates directly correspond to probabilities, making conversion straightforward, computationally efficient.

---

> ### Author Response · Authors · 2024-11-23
>
> ## Question 1:  The explanation of distribution-based similarity correlations and SSSA's spike-driven characteristics.
> **A:**As you noted, the computation of distribution-based correlations can be interpreted as evaluating the correlations between the firing rates of vectors in Q and K. This method aligns well with the principles of frequency encoding. Moreover,  the SSSA-V1 model faces challenges with the outer product of integer vectors. To address this, we develope SSSA-V2, ensuring spike-driven characteristics. In the SSSA-V2 version, the calculation formulas are as followed:
> $$
> \mathcal{Q}=\sum_{i=1}^{D} Q(i,j),\quad \mathcal{K}^T = \sum_{i=1}^{D}K^T(i,j),\quad \mathbf{L} = \begin{bmatrix}
> 1 \\\\
> 1 \\\\
> 1 \\\\
> \vdots \\\\
> 1
> \end{bmatrix} \quad \mathcal{P}atch = \mathcal{Q} (\mathcal{K}^T \times L).
> $$
> $(\mathcal{K}^T \times L)$ is a integer scalar value. To avoid this integer multiplication $\mathcal{Q} (\mathcal{K}^T \times L)$，We treat $(\mathcal{K}^T \times L)$ as a learnable scaling factor $\alpha$, which is applied to the $\textbf{V}\_{th}$ of the saccadic neuron. Therefore, combining the asynchronous decoupling approach as above, the formula for SSSA-V2 in inference can be expressed as follows:
> $$
> \left\\{
> \begin{array}{ll}
> H[t] = \mathcal{Q}[t], \\\\
> S[t] = \Theta \left(\textbf{H}[t]- \frac{1}{\alpha} \left(\textbf{M}\_{w}^{-1} \textbf{V}\_{th} \right)[t]\right).
> \end{array}
> \right.
> $$
>
> In this manner, $\frac{1}{\alpha} \left(\textbf{M}\_{w}^{-1} \textbf{V}\_{th}\right)[t]$ is the $\textbf{V}_{th}$ for saccadic neurons at timestep $t$, thus ensuring the fully spike-driven nature of SNN-ViT.
>
> ## Question 2: Would V1 perform better when floating-point computations are involved?
> **A:** Our subsequent experiments are conducted on SSSA-V2.  In the training process, SSSA-V2 and V1 exhibit a linear scaling relationship with virtually no performance loss. The distinction between them emerges during the inference process, where V2 integrates $(\mathcal{K}^T \times L)$ into the threshold $V_{th}$ of the saccadic neurons as a constant value, whereas in V1, it is variable. We also conduct experiments that show minimal performance disparity between the two versions, and SSSA-V2 successfully reduces the computational complexity from $\mathcal{O}\left(N^2\right)$to $\mathcal{O}\left(D\right)$.

---

### Official Review · Reviewer_Rjtr · 2024-11-02

**Soundness:** 2
**Presentation:** 3
**Contribution:** 2
**Rating:** 5
**Confidence:** 4

**Summary:**

The author of this paper conducted research on the architecture of introducing Vision Transformers into spiking neural networks (SNNs). By analyzing the distribution of Query and Key in vanilla self-attention mechanisms and existing self-attention mechanisms in SNNs, the author pointed out the shortcomings of current SNN+Transformer approaches. To address these shortcomings, the author introduced the method of Sacadic Spike Self-Attention and designed the SNN-ViT network framework based on this method. The performance of the proposed approach was validated on various tasks and datasets.

**Strengths:**

The author has a deep understanding of the existing SNN+Transformer work and has summarized the common shortcomings of these studies.

**Weaknesses:**

1. The theoretical derivation is insufficient; the proof related to the concept of "equivalence" provided in the paper is not included in the appendix, and there are incomplete formulas in some places, such as Eq. (7).
2. The understanding of SNNs is not profound, including the basic processes of biological spiking neurons.
3. There are insufficient ablation experiments to adequately support the effectiveness of the various new methods proposed in the paper.
4. There is a lack of understanding of the cited references, especially papers closely related to the methods in this study, and there is no comparison or discussion of these methods with the approaches presented in this paper.

**Questions:**

1. The author points out in the paper that the incompatibility of LayerNorm with the spike-driven characteristics of SNNs leads to a significant difference in the distribution of Query and Key, resulting in performance degradation. However, in reality, 1) many SNN studies have replaced LayerNorm with BatchNorm to allow BN to be merged into Conv during inference, thus maintaining the spike-driven characteristics, and 2) the LayerNorm operation should be mergeable into Linear during inference. In this case, is the author's analysis of the distribution of Query and Key unfair? Because there is no normalization applied to the Query and Key in this context and the normalization operation itself can be incorporated into the SNN.
2. In section 3.2, the paper points out that previous work did not introduce temporal information modeling in the design of self-attention operators. However, it is well known that SNN inherently model the temporal features of inputs by the spiking neuron, and often involve the insertion of spiking neurons across many layers of the network. Therefore, why does the author insist on adding an additional temporal modeling component to the self-attention operator?
3. In section 3.2, the author states, "However, due to the reset and decay mechanism, the residual membrane potential cannot sustain long-range dependencies, resulting in a significant loss of historical information." This assertion is inherently incorrect, as biological neurons include decay and reset mechanisms, which actually enable spiking neurons to avoid excessive firing and effectively retain memory. The author's outright dismissal of the contributions of decay and reset operations to neurons raises doubts about whether the author truly understand spiking neural networks.
4. I believe that the mechanism of the Saccadic Neuron inserted by the author in the self-attention mechanism bears a high similarity to the Parallel Spiking Neuron proposed in [1], especially in Equation (8). While I noticed that the author referenced this literature in the paper, there was no mention of the content regarding PSN in the section where Saccadic Neuron is introduced. I would like the author to address what the relationship and differences are between Saccadic Neuron and PSN. Why not use PSN directly?
5. The author presents the so-called equivalence between training and inference in Eq.(8), but I am skeptical about this equivalence. I hope the author can provide a detailed proof of the equivalence between the two processes.
6. The author introduces a new modeling method and neuron model in the SSSA module. While this may reduce complexity to some extent, it also seems to introduce MAC operations, specifically reflected in the integer vector outer product in SSSA-V1 and the Hadamard product operation in SSSA-V2.
7. Table 2 presents a performance comparison on ImageNet-1k. When compared with SpikingResformer, it can be seen that at 13.7M vs. 17.8M, the energy consumption is 14.28mJ and 3.37mJ, with corresponding performances of 74.66% and 75.95%; at 30.4M vs. 35.5M, the energy consumption is 20.83mJ and 5.46mJ, with performances of 76.87% and 77.24%. This indicates that there is an advantage in performance only when the model scale is increased, while there seems to be no significant advantage in smaller networks. Additionally, the claimed contribution of reduced computational complexity appears to remain at a theoretical level, as there is no advantage from the perspective of energy consumption.
8. In the design of the ablation experiments, the paper only provides experimental results for the SSSA and GL-SPS modules, without analyzing the contribution of the internal components of the proposed SSSA to performance. For example, it does not discuss the performance of SSSA-V1 or the effect of replacing the Saccadic Neuron with a standard LIF neuron.
[1] Fang, Wei, Zhaofei Yu, Zhaokun Zhou, Ding Chen, Yanqi Chen, Zhengyu Ma, Timothée Masquelier, and Yonghong Tian. "Parallel spiking neurons with high efficiency and ability to learn long-term dependencies." Advances in Neural Information Processing Systems 36 (2024).

---

> ### Author Response · Authors · 2024-11-23
>
> Thank you for your insightful feedbacks. We will address each of your points in turn.
> ## Question 1：Analysis of the distribution of Query and Key is unfair?
> **A:** We apologize for the confusing statements regarding the mismatch between LayerNorm and spike trains. As you pointed out, normalization operations can be integrated with convolutional and linear layers in SNNs. We clarify that the membrane potentials of our $\mathcal{Q}$ and $\mathcal{K}$ spike trains are computed through batch normalization. The formulations for our $\mathcal{Q}$ and $\mathcal{K}$ are as follows:
> $$
> \mathcal{Q} = \mathcal{SN}(BN(Conv(X))), \quad K = \mathcal{SN}(BN(Conv(X)))
> $$
> where $BN$ represents batch normalization and $Conv$ denotes the convolution process. Therefore, the comparison of the magnitudes of Q and K in SNN and ANN is fair. Through comparative analysis as Fig.1, we point that differences in the magnitudes of $\mathcal{Q}$ and $\mathcal{K}$ are attributable to the discrete activation characteristics of $\mathcal{SN}$. In the revised manuscript, we have clarified the processing for Q and K, as well as Fig. 2.
>
> ## Question 2 ： Why does the author insist on adding an additional temporal modeling component to the self-attention?
> **A:** As you mentioned, SNNs possess neural dynamics capable of processing spatio-temporal information. However, many studies[1-3] propose that designing temporal components (TC) can further enhance the performance ceiling of SNNs. For instance, Yao et al.[1] introduced a temporal attention module into SNNs, allowing the model to adaptively assign importance to different time steps. Shen et al.[2] proposed learnable decay factor for the spiking neurons at each timestep.
>
> |   Dataset   |      Method       | Param (M) |            Complexity             |      Acc (%)     |
> | :---------: | :---------------: | :-------: | :-------------------------------: | :--------------: |
> |   CIFAR10   |  **SSSA(Ours)**   | **5.52M** |   $\mathcal{O}\left(D\right)$     |     96.12%       |
> |             | SSSA( without TC) |   5.76M   |  $\mathcal{O}\left(N^2 D\right)$  |      95.20%      |
> |  CIFAR100   |  **SSSA(Ours)**   | **5.52M** |    $\mathcal{O}\left(D\right)$    |     79.60%       |
> |             | SSSA( without TC) |   5.76M   | $\mathcal{O}\left(N^2 D\right)$   |      78.38%      |
>
> Moreover, we conduct performance evaluations of our SSSA on the CIRAF100 dataset both with and without saccadic temporal interactions. As illustrated in the table above, it reveals that our SSSA achieves an approximate 1.2% performance improvement under identical structural conditions.
>
> Reference:
>
> [1] Yao, M., Gao, H., Zhao, G., Wang, D., Lin, Y., Yang, Z., & Li, G. Temporal-wise attention spiking neural networks for event streams classification. ICCV(2021).
>
> [2] Yin, B., Corradi, F., & Bohté, S. M. Accurate and efficient time-domain classification with adaptive spiking recurrent neural networks. Nature Machine Intelligence, 3(10), 905-913, 2021.
>
> [3] Zhu, R. J., Zhang, M., Zhao, Q., Deng, H., Duan, Y., & Deng, L. J. (2024). Tcja-snn: Temporal-channel joint attention for spiking neural networks. _IEEE Transactions on Neural Networks and Learning Systems_.
>
> ## Question 3： Reset and decay mechanism in saccadic neurons？
> **A:** As you highlighted, reset and decay mechanisms in spiking neurons are crucial for preventing excessive neuron firing and preserving memory capability. However, numerous studies[1-2]  also tried to remove the reset and decay mechanisms during training, achieving significant performance on many long-sequence tasks. To demonstrate that our saccadic neurons also possess sparse and have better performance compared to LIF, we conducted experiments on sequence tasks.
>
> | Dataset  | Model  | Fire_Ratio  |     Acc.(%)      |
> | :------: | :----: | :---------: | :--------------: |
> | s-mnist  |  LIF   |    25.3%    | $98.70 \pm 0.25$ |
> | s-mnist  | Ours   |   22.3%     | $99.20\pm 0.05$  |
> | ps-mnist |  LIF   |   26.3%     | $94.30 \pm 0.35$ |
> | ps-mnist | Ours   |   21.5%     | $98.20\pm 0.20$  |
> - **Better performance**: In sequence tasks, models based on saccadic neurons demonstrate significantly improved accuracy and outperform those based on LIF neurons. This indicates that saccadic neurons better preserve long-term dependencies and effectively utilize historical information.
> - **Sparsity**: As shown above, the average firing rates of saccadic neurons are comparable to those of LIF neurons, maintaining network sparsity.
>
> Thus, our saccadic neurons possess both better performance and Sparsity characteristics.
>
> Reference:
>
> [1] Su Q, Mei S, Xing X, et al. SNN-BERT: Training-efficient Spiking Neural Networks for energy-efficient BERT[J]. Neural Networks, 2024, 180: 106630.
>
> [2] Fang, W., Yu, Z., Zhou, Z., Chen, D., Chen, Y., Ma, Z., ... & Tian, Y. (2024). Parallel spiking neurons with high efficiency and ability to learn long-term dependencies. _Advances in Neural Information Processing Systems_, _36_.

---

> > ### Author Response · Authors · 2024-11-23
> >
> > ## Question 4 : The difference between PSN and saccadic neurons.
> > **A:** Our saccadic neurons exhibit significant structural differences from PSN neurons. We will thoroughly explore these differences by analyzing both the training and inference processes.
> >
> > | Dataset   |     Model     | SOPs in Inference (K) |     Acc.(%)      |
> > | :-------: | :-----------: | :-----------------: | :--------------: |
> > |  s-mnist  | PSN（window=4） |       338.18        | $98.90 \pm 0.10$ |
> > |  s-mnist  |     Ours      |        84.55        | $99.20\pm 0.05$  |
> > | ps-mnist  | PSN（window=4） |       338.18        | $97.00 \pm 0.24$ |
> > | ps-mnist  |     Ours      |        84.55        | $98.20\pm 0.20$  |
> >
> > **Training process:** The PSN neuron focuses on information from adjacent timesteps, while the saccadic neuron considers information from all previous timesteps. This mechanism allows the saccadic neuron to allocate input weights more effectively.
> >
> > **Inference process:** The PSN model requires computation of all information within the current window(2/4/8 timesteps), whereas the Saccadic Neuron focuses solely on the input at the current moment utilizing decoupling mechanisms. As demonstrated in Eq.8, we can integrate this information into the threshold to ensure energy efficiency.
> >
> > Additionally, we tested the performance of our saccadic neurons against PSN across sequence tasks. The results indicate that our saccadic neuron achieves superior performance with less SOPs costs during inference.
> >
> > ## Question 5: Detailed proof of the equivalence between training and inference processes.
> > **A:** We sincerely apologize for any confusion caused by Equation 8. We will provide detailed proof of the equivalence between the saccadic neurons during the training and inference processes.
> > Firstly, the dynamic of saccadic neurons in Training and Inference phase can be described as follows:
> > $$
> > Training
> > \left\\{
> > \begin{array}{l}
> > \textbf{H} =  \textbf{M}_w \mathcal{P}atch, \\\\
> > \textbf{S} =  \left\\{
> > \begin{array}{l}
> > 1, & \text{if }\textbf{H} \geq \textbf{V}\_{th}, \\\\
> > 0, & otherwise.
> > \end{array}
> > \right.
> > \end{array}
> > \right. \quad
> > Inference
> > \left\\{
> > \begin{array}{l}
> > \textbf{H}[t] =   \mathcal{P}atch[t], \\\\
> > \textbf{S} =  \left\\{
> > \begin{array}{l}
> > 1, & \text{if }\textbf{H}[t] \geq \textbf{M}_w^{-1}\textbf{V}'\_{th}[t], \\\\
> > 0, & otherwise.
> > \end{array}
> > \right.
> > \end{array}
> > \right.
> > $$
> >
> > $\mathcal{P}atch \in \mathbb{R}^{T \times N}$ represents the spatial importance of different regions, $\textbf{M}\_w$ is a lower triangular matrix, which can represent $\textbf{M}\_w = \left\(
> > \begin{matrix}
> > w_{11}&\cdots&0\\\\
> > \vdots&\ddots&\vdots\\\\
> > w_{n1}&\cdots&w_{nn}\\\\
> > \end{matrix}
> > \right\)$.Additionally, The bolded letters $\textbf{H}$ and $\textbf{V}\_{th}$ represent the membrane potential and threshold that include temporal dimensions, whereas the non-bold letters denote $H[t]$ and $V\_{th}[t]$ at a specific timestep.
> >
> >
> > **Proof:** Training and Inference are equivalent under the Condition that $\textbf{M}_w$​ is Invertible.
> >
> > **1.Lemma:** A lower triangular matrix is invertible if and only if all its diagonal elements are nonzero.
> >
> > The diagonal elements of $\textbf{M}\_w$ represent the proportional weights of $\mathcal{P}atch$ at the current time step, and we ensure they are nonzero during training. Therefore, $\textbf{det}\left( \textbf{M}\_{w} \right) \neq 0$, then $\textbf{M}\_w$​ is Invertible.
> >
> > If $\textbf{M}_w$ is invertible, we can transform the training by applying $\textbf{M}_w^{-1}$ to both sides of the equation for training process:
> > $$
> > \textbf{M}_w^{-1}\textbf{H} = \textbf{M}_w^{-1}(\textbf{M}_w \mathcal{P}atch) = \mathcal{P}atch
> > $$
> > $$
> > \textbf{S} =
> > \begin{cases}
> > 1, & \text{if } \mathcal{P}atch \geq \textbf{M}\_w^{-1}\textbf{V}\_{th}, \\\\
> > 0, & \text{otherwise}.
> > \end{cases}
> > $$
> > Therefore, through this transformation, the training process is completely equivalent to the inference formula.
> > In summary, asynchronous decoupling is achieved by integrating $\textbf{M}_w$ learned during training into the $\textbf{V}\_{th}$ at each moment, creating dynamic thresholds. This ensures that the network can consider historical information while only needing to compute inputs at the current moment. thereby enabling asynchronous inference.

---

> > > ### Author Response · Authors · 2024-11-23
> > >
> > > ## Question 6： Details of SSSA-V2 for spike-driven characteristics.
> > > **A:**As you mentioned, our SSSA-V1 indeed involves integer multiplication during the inference process. To further optimize this, we introduce scaling mapping version SSSA-V2. I will provide a detailed analysis of how SSSA-V2 ensures spike--driven characteristics during inference. The calculation formula for this part can be expressed as follows:
> > > $$
> > > \mathcal{Q}=\sum_{i=1}^{D} Q(i,j),\quad \mathcal{K}^T = \sum_{i=1}^{D}K^T(i,j),\quad \mathbf{L} = \begin{bmatrix}
> > > 1 \\\\
> > > 1 \\\\
> > > 1 \\\\
> > > \vdots \\\\
> > > 1
> > > \end{bmatrix} \quad \mathcal{P}atch = \mathcal{Q} (\mathcal{K}^T \times L).
> > > $$
> > > $(\mathcal{K}^T \times L)$ is a integer scalar value. To avoid this integer multiplication $\mathcal{Q} (\mathcal{K}^T \times L)$，We treat $(\mathcal{K}^T \times L)$ as a learnable scaling factor $\alpha$, which is applied to the $\textbf{V}\_{th}$ of the saccadic neuron. Therefore, combining the asynchronous decoupling approach as above, the formula for SSSA-V2 in inference can be expressed as follows:
> > > $$
> > > \left\\{
> > > \begin{array}{ll}
> > > H[t] = \mathcal{Q}[t], \\\\
> > > S[t] = \Theta \left(\textbf{H}[t]- \frac{1}{\alpha} \left(\textbf{M}\_{w}^{-1} \textbf{V}\_{th} \right)[t]\right).
> > > \end{array}
> > > \right.
> > > $$
> > > In this manner, $\frac{1}{\alpha} \left(\textbf{M}\_{w}^{-1} \textbf{V}\_{th}\right)[t]$ is the $\textbf{V}_{th}$ for saccadic neurons at timestep $t$, thus ensuring the fully spike-driven nature of SNN-ViT.
> > >
> > > ## Question 7：Details of performance and energy consumption.
> > > **A:** As you noted, our SNN-VIT exhibits more pronounced performance improvements in larger model sizes. We also considered the reasons. In smaller networks, self-attention modules largely function as Token Mixers as proposed by MetaFormer[1]. Thus, even ineffective correlation computations can still perform well. While in larger scales, effective spatial correlation computations perform clear advantages. Additionally, as shown in the table below, we have compiled energy consumption comparisons for various SSA configurations under conditions of Head=8, D=512, N=196, T=4. Notably, our SSSA demonstrates significantly lower SOPs and reduced computational energy consumption. Therefore, in the self-attention computation module, our SSSA exhibits a clear advantage in energy efficiency.
> > >
> > > | Spike self-attention |       Complexity       | SOP in Attention | Energy (uJ) |
> > > | :------------------: | :--------------------: | :--------------: | :---------: |
> > > |         SSA          |  $\mathcal{O}(N^2D)$   |      39.34M      |    35.04    |
> > > |         STSA         | $\mathcal{O}(T^2N^2D)$ |     178.68M      |   160.81    |
> > > |        SDSA-4        |  $\mathcal{O}(ND^2)$   |      12.58M      |    11.32    |
> > > |         SSSA         |    $\mathcal{O}(D)$    |      1.21M       |    1.089    |
> > >
> > > Reference:
> > >
> > > [1] Yu, W., Luo, M., Zhou, P., Si, C., Zhou, Y., Wang, X., ... & Yan, S. (2022). Metaformer is actually what you need for vision. In _Proceedings of the IEEE/CVF conference on computer vision and pattern recognition_ (pp. 10819-10829).
> > >
> > > ## Question 8: Further ablation experiments for SSSA.
> > >
> > > **A**: Thank you for your valuable suggestions.  We add ablation studies on the two key components of the SSSA module: (1) Replacing **distribution-based spatial similarity** with traditional **Dot Product (DP)** similarity.(2) Replacing **saccadic neurons** with LIF neurons.  Finally, we also compare the performance of versions V1 and V2. Experiments are performed on the CIFAR100 dataset, and the results are presented in the following Table.
> > >
> > > |  model   | Param (M) |           Complexity            |   Acc (%)     |
> > > | :------: | :-------: | :-----------------------------: | :-----------: |
> > > | Baseline |   5.76M   | $\mathcal{O}\left(N^2 D\right)$ |     76.95     |
> > > | SSSA+DP  |   5.52M   | $\mathcal{O}\left(N^2 D\right)$ |     77.12     |
> > > | SSSA+LIF |   5.52M   |   $\mathcal{O}\left(D\right)$   |     78.84     |
> > > | SSSA-V1  | **5.52M** |  $\mathcal{O}\left(N^2\right)$  |     79.71     |
> > > | SSSA-V2  | **5.52M** |   $\mathcal{O}\left(D\right)$   |     79.60     |
> > >
> > > The SSSA+DP shows almost no performance improvement compared to the baseline. This underscores that effective spatial similarity computation is the foundation for subsequent saccadic interactions.  Then, substituting saccadic neurons with LIF neurons led to an approximate 0.8% decrease in performance relative to SSSA.  This demonstrates that saccadic interactions can indeed enhance performance.  Finally, while there is virtually no performance disparity between V1 and V2, the computational complexity of V2 is only $\mathcal{O}\left(D\right)$.  In summary, our SSSA-V2 module achieves an optimal trade-off between computational complexity and performance. (Appendix.E)

---

### Official Review · Reviewer_CNSS · 2024-11-03

**Soundness:** 3
**Presentation:** 2
**Contribution:** 3
**Rating:** 6
**Confidence:** 4

**Summary:**

The paper presents a novel SNN-ViT incorporating a Saccadic Spike Self-Attention (SSSA) mechanism designed to leverage the spatio-temporal characteristics of SNNs for vision tasks. The authors identify key limitations in traditional self-attention for SNNs and introduce methods to enhance spatial relevance assessment and temporal dynamics.

**Strengths:**

1. The experiment results show that the provided models can outperform previous methods on various tasks.

2. The SNN-ViT’s linear computational complexity, achieved through SSSA-V2, enhances its viability for energy-critical applications and distinguishes it from previous methods that rely heavily on expensive MAC operations.

3. This paper is well written with a detailed appendix to understand their theory.

**Weaknesses:**

1. The attention heatmaps in Figures 5 and 7 appear noisy and lack interpretability compared to the original ViT. It is challenging to correlate these attention patterns with the final detection results. Additional explanation of these heatmaps would strengthen the paper’s presentation and the interpretability of the results.

2. The paper lacks ablation studies on the design of the "patch" mechanism within the saccadic interaction module. Would performance improve if all values in the QK matrices were used, even if this increases computational complexity? Additionally, it would be helpful to analyze whether the "patch" design enhances generalizability across various datasets or tasks.

**Questions:**

1. Figure 1 lacks a clear explanation. In this figure, I assume the y-axis represents the number of samples and the x-axis represents a numerical magnitude, but this is not explicitly stated. Please add axis labels with units in the revised manuscript. Additionally, these numbers were obtained through simulation. Will there be simulation-reality gap? Would real datasets yield similar distributions or statistics? Does different dataset impact the statistics?

2. Could you provide a reference or further clarification for the statement in lines 277–278? Adding a source or further explanation here would help substantiate this claim.

---

> ### Author Response · Authors · 2024-11-23
>
> We greatly appreciate your recognition of the innovative aspects and motivation of our work. In response to the weaknesses and suggestions you have raised, we will provide further **detailed explanations**:
>
> ## Weakness 1: Additional explanation of these heatmaps.
>
> **A**: Thank you for your suggestions. Our heatmap display is noisy mainly for two reasons. Firstly, the SSSA module, driven by spikes, operates in two states at each timestep: focus or ignore. This binary operation makes its behavior less smooth compared to full-precision ViT networks but ensures highly sparse. Second, because our display shows multi-target remote sensing object detection, the network focuses on all visual areas close to the ten types of targets. Consequently, on the NWPU-10 dataset, the heatmaps also show attention to areas outside the target boxes. Following your advice, we have added visualizations for a single-target detection dataset SSDD. This updated visualization is included in the new manuscript.(Appendix.G)
>
> ## Weakness 2: Lacking ablation studies on the SSSA.
>
> A: Thank you for your valuable suggestions. In the revised manuscript, we design additional ablation experiments to address your concerns. Specifically, we evaluate the performance of the method that **uses all QK values**. Additionally, we evaluate the performance of our method on the CIFAR10/100 and CIFAR10DVS datasets to demonstrate its generalizability.
>
> |   Dataset   |     Method     | Param (M) |           Complexity             |     Acc (%)     |
> | :---------: | :------------: | :-------: | :------------------------------: | :-------------: |
> |   CIFAR10   | **SSSA(Ours)** | **5.52M** |   $\mathcal{O}\left(D\right)$    |     96.12%      |
> |             |    Full-QK     |   5.76M   | $\mathcal{O}\left(N^2 D\right)$  | 96.20%  |
> |  CIFAR100   | **SSSA(Ours)** | **5.52M** |   $\mathcal{O}\left(D\right)$    |     79.60%      |
> |             |    Full-QK     |   5.76M   | $\mathcal{O}\left(N^2 D\right)$  | 79.78% |
> | CIFAR10-DVS | **SSSA(Ours)** | **1.52M** |   $\mathcal{O}\left(D\right)$    |     82.30%      |
> |             |    Full-QK     |   1.76M   | $\mathcal{O}\left(N^2 D\right)$  | 82.35%  |
>
> As demonstrated in the table, "QK-based" denotes the method that **utilizes all QK values**.  For various datasets, the "QK-based" method increases the computational complexity to $\mathcal{O}(N^2D)$ but yields only limited performance improvements. Therefore, our SSSA module achieves an optimal trade-off between computational complexity and performance.
>
> Moreover, the "Patch" design yields high performance and lower computation complexity across various datasets, demonstrating its generalizability. (Appenidx.E)
>
> ## Questions 1: The explanation of Figure 1
>
> **A**: Thank you for your suggestion. In the revised manuscript, we have added X and Y labels to Figure 1 and included details about the network architecture in the figure caption.  The X-axis represents the vector magnitudes, and the Y-axis indicates the number of samples. As per your recommendation, our data are not simulated but collected from training an ANN-based ViT model and a Spikformer on the CIFAR100 dataset for 300 epochs, both configured with $D = 384$ and 12 heads. Additionally, following your advice, we observe abnormal distributions of $\mathcal{Q}$ and $\mathcal{K}$  vector in SNNs across multiple datasets, indicating that different datasets do not affect the statistical results.（Fig.1）
>
> ## Questions 2: Further clarification for the statement in lines 277–278? Adding a source or further explanation here would help substantiate this claim.
>
> **A**: Your suggestions have been immensely helpful in enhancing the quality of our manuscript. Following your advice, we have added Appendix C to detail this part. Numerous neuroscience findings[1-3] confirm that the eyes do not acquire all details of a scene simultaneously. Instead, attention is focused on specific regions of interest (ROIs) through a series of rapid eye movements called saccades. Each saccade lasts for a very brief period—typically only tens of milliseconds—allowing the retina's high-resolution area to sequentially align with different visual targets. This dynamic eye movement mechanism enables the visual system to process information efficiently by avoiding redundant processing of the entire visual scene. (Appenidx.C)
>
> References:
>
> [1] Melcher, David, and M. Concetta Morrone. "Spatiotopic temporal integration of visual motion across saccadic eye movements." Nature neuroscience 6.8 (2003): 877-881.
>
> [2] Binda, Paola, and Maria Concetta Morrone. "Vision during saccadic eye movements." Annual review of vision science 4.1 (2018): 193-213.
>
> [3] Guadron, Leslie, A. John van Opstal, and Jeroen Goossens. "Speed-accuracy tradeoffs influence the main sequence of saccadic eye movements." Scientific reports 12.1 (2022): 5262.

---

### Public Comment · ~jinfan_cheng1 · 2025-08-08

Dear authors，
Great work.I find your paper quite interesting.But I am a bit confused about Equation[8].There was mentioned that M_w is for facilitating efficient temporal interactions.However, given that Mw has a size of n×n, it appears to represent interactions between neurons within one single time step.
I apologize if this question seems basic, but could you please clarify my confusion?

---

> ### Public Comment · ~Shuai_Wang23 · 2025-08-08
>
> Thank you for your recognition of our work. Regarding the M_w you mentioned, its dimension is not N×N but rather a T×T lower triangular matrix, which is exclusively responsible for temporal information interaction. In Equation 8, both H and Patch have dimensions of T×N; therefore, M_w must be T×T to ensure proper matrix multiplication. We hope this clarification helps you. Thank you again for your acknowledgment of our work, and we wish you success in your endeavors.

---

> > ### Public Comment · ~jinfan_cheng1 · 2025-08-11
> >
> > Dear authors,
> > Thank you so much for your kind response.I truly appreciate you taking the time to address my query and provide such a clear explanation regarding the dimension of M_w.

---

### Meta-Review · Area_Chair_aZzR · 2024-12-22

**Metareview:**

This paper introduces a novel Vision Transformers (ViTs) framework based on spiking neural networks (SNNs) and proposes a bio-inspired Saccadic Spike Self-Attention (SSSA) method to improve the performance of SNNs in visual tasks. To address the compatibility issue between the traditional self-attention mechanism and SNNs, the authors propose techiniques to enhance the performance of temporal dynamics. Overall, this paper proposes an SNN-ViT scheme that can achieve linear computational complexity and demonstrates its potential in neuromorphic applications. After the rebuttal stage, 5 reviewers gave ratings of 5, 6, 6, 8, 8. Therefore, I recommend this paper to be accepted as a poster paper by ICLR.

**Additional Comments On Reviewer Discussion:**

The reviewers mainly acknowledged that this paper provides an efficient SNN-ViT model and shows excellent experimental performance. The computational complexity of SNN-ViT has been effectively optimized. However, the reviewers also raised several criticisms, mainly focusing on the model interpretability and the shortcomings of ablation experiments (reviewers CNSS, Rjtr). In the rebuttal, the author further explained the model design and experimental details in response to the reviewer's comments. During the review stage, some minor problems were found in this paper. For example, the figure labels, formula explanation, etc. The author has responded to these problems in the rebuttal. I suggest that the author further supplement the relevant explanations, analysis and experimental data according to the reviewer's comments.

---

### Decision · Program_Chairs · 2025-01-22

Accept (Poster)